# MIND THE GAP:
# A SPECTRAL ANALYSIS OF RANK COLLAPSE
# AND SIGNAL PROPAGATION IN TRANSFORMERS

## ABSTRACT

Attention layers are the core component of transformers, the current state-of-the-art neural network architecture. However, softmax-based attention causes transformers to be more challenging to train. Even *at initialisation*, the propagation of signals and gradients through the random network can be pathological, resulting in known issues such as (i) vanishing/exploding gradients and (ii) *rank collapse*, i.e. when all tokens converge to a single representation *with depth*. This paper examines signal propagation in *attention-only* transformers from a random matrix perspective, illuminating the origin of such issues, as well as unveiling a new phenomenon—(iii) rank collapse *in width*. Modelling softmax-based attention at initialisation with Random Markov matrices, our theoretical analysis reveals that a *spectral gap* between the two largest singular values of the attention matrix causes (iii), which, in turn, exacerbates (i) and (ii). Building on this insight, we propose a novel, yet simple, practical solution to resolve rank collapse in width by removing the spectral gap. Moreover, we validate our findings and discuss the training benefits of the proposed fix through experiments[1] that also motivate a revision of some of the default parameter scaling. Our attention model accurately describes the standard key-query attention in a single-layer transformer, making this work a significant first step towards a better understanding of the initialisation dynamics in the multi-layer case.

## 1 INTRODUCTION

Transformers Vaswani et al. (2017) have emerged as the dominant architecture in machine learning, achieving remarkable success across various domains, particularly in natural language processing and computer vision, largely due to their defining feature: the self-attention mechanism Bahdanau et al. (2016). However, despite their empirical success, transformers are often plagued by training instability and high sensitivity to numerous hyperparameters, which require careful tuning. This challenge has motivated recent efforts to establish a theoretical framework for understanding transformer architectures, even in their most basic forms, to ensure reliable information flow through deeper layers and facilitate training.

The purpose of this work is to analyse signal propagation in softmax-based attention layers *at initialisation*, i.e. with randomly initialised model parameters. While the issues of rank collapse (in depth) and vanishing/exploding gradients have been previously identified in transformers at initialisation Dong et al. (2021); Noci et al. (2022), our work formalises these findings and uncovers an additional phenomenon—rank collapse *in width*—due to the use of softmax in the self-attention mechanism. Rank collapse in width has not been identified in the existing literature nor been recognised as a catalyst for rank collapse along depth. By leveraging spectral properties of the random matrices formed by the model's parameters, we reveal the emergence of a *spectral gap* between the two largest singular values of the attention matrix, which drives rank collapse in width and further accelerates rank collapse in depth. Moreover, we propose a provably effective remedy for the spectral gap, a solution that naturally arises when the problem is viewed through a spectral lens. To the best of our knowledge, a spectral analysis of signal propagation has yet not been undertaken in the context of transformers.

---

[1]Our code is available at https://shorturl.at/0zk8q.

Let us consider the eigenvalues of an attention matrix. Since the rows sum to 1, there is an eigenvalue of 1 corresponding to the all-ones vector. Under certain conditions, the other eigenvalues shrink in size as the matrix dimension increases, resulting in a widening gap between the largest eigenvalue (which is 1) and the diminishing bulk of eigenvalues; see Figure 1. The successive multiplication of such matrices at each layer increasingly favours a specific direction—the one aligned with the dominant eigenvector of the attention matrix—over the others. This leads to a distortion in the geometry of the input training data, exemplified by the phenomenon of rank collapse. A natural solution is then to project out this troublesome direction from all attention matrices to enable a more balanced signal propagation. This intuitive idea is central to our rigorous analysis of a simple transformer, from which we draw insights to introduce a slightly modified attention layer that proves advantageous even when incorporated into more complex architectures.

**Spectra of random matrices.** Throughout this paper, we consider random matrices (of different distributions) in the *large width* limit and describe them through their *limiting* spectral properties. In the context of transformers, we mean by "large width" that both the number of tokens and the embedding dimension(s) are large—an assumption typically satisfied in practice. For certain classes of random matrices, the overall behaviour of eigenvalues/singular values becomes remarkably predictable as the matrix size increases, despite the randomness of individual entries. If $\mathbf{M}_n \in \mathbb{R}^{n \times n}$ are random matrices with eigenvalues and singular values denoted by $\{\lambda_i(\mathbf{M}_n)\}$ and $\{s_i(\mathbf{M}_n)\}$, respectively, the histograms of the $n$ eigenvalues/singular values

$$\mu_{\mathbf{M}_n} := \frac{1}{n} \sum_{i=1}^{n} \delta_{\lambda_i(\mathbf{M}_n)}, \quad \nu_{\mathbf{M}_n} := \frac{1}{n} \sum_{i=1}^{n} \delta_{s_i(\mathbf{M}_n)},$$

converge, in many interesting cases, to deterministic distributions $\mu$ and $\nu$, known as the *limiting eigenvalue/singular value distribution* of $\mathbf{M}_n$. Additionally, the *largest* eigenvalue/singular value of random matrices is often studied in its own right. Our analysis builds on several established results concerning both the limiting distribution of the eigenvalues/singular values (the "bulk" of the spectrum) and the behaviour of the largest eigenvalue/singular value (the "edge" of the spectrum). In particular, we focus on two classes of random matrices: Gaussian and Markov, that respectively model the *value* and *attention* matrices in our transformer model (3) at initialisation.

**Free probability.** The theory of free probability studies "non-commuting random variables" such as random matrices (see Mingo & Speicher (2017) for a textbook introduction). Pioneered by Pennington et al. (2017; 2018), the theory has found powerful applications in the analysis of large random neural networks. Notably, it provides tools to characterise the singular value distribution of sums or products of random matrices. Loosely speaking, "freeness" plays the same role for random matrices as independence does for (scalar) random variables. Freeness allows us to compute the limiting spectral density of a product $\mathbf{M}_n \mathbf{M}'_n$ from the limiting spectral densities of $\mathbf{M}_n$ and $\mathbf{M}'_n$, just as independence enables the computation, for instance, of the moments of $ZZ'$, given those of $Z$ and $Z'$. Specifically, if $\nu_{\mathbf{M}_n} \to \nu$, $\nu_{\mathbf{M}'_n} \to \nu'$, and $\mathbf{M}_n$ and $\mathbf{M}'_n$ are *asymptotically* free, then

$$\nu_{\mathbf{M}_n \mathbf{M}'_n} \xrightarrow{n \to \infty} \nu \boxtimes \nu',$$

where $\boxtimes$ denotes an operation called *free multiplicative convolution*.

## 1.1 ATTENTION AT INITIALISATION

We model the attention mechanism at initialisation by a random matrix $\mathbf{A}_\ell$ with non-negative entries $(\mathbf{A}_\ell)_{i,j} \geq 0$ and normalised rows, i.e. $\sum_j (\mathbf{A}_\ell)_{i,j} = 1$, as if it were generated by a row-wise application of softmax. As we will demonstrate, this model functions as a helpful abstraction that offers insights into the causes of rank collapse. More specifically, we consider $\mathbf{A}_\ell$ to be a Random Markov matrix, as defined in Bordenave et al. (2011).

**Definition 1.1** (Random Markov matrix). *Let $Z_{i,j}$ be i.i.d. non-negative random variables with positive mean $m := \mathbb{E}(Z_{1,1}) > 0$ and variance $\sigma^2 := \mathrm{Var}(Z_{1,1}) > 0$ as well as finite fourth moment $\mathbb{E}(Z_{1,1}^4) < \infty$. Let $\mathbf{A} \in \mathbb{R}^{T \times T}$ be its row-normalised version, i.e.,*

$$\mathbf{A}_{i,j} := \frac{Z_{i,j}}{\sum_{j=1}^{T} Z_{i,j}}. \tag{1}$$

*We call $\mathbf{A}$ a Random Markov matrix.*

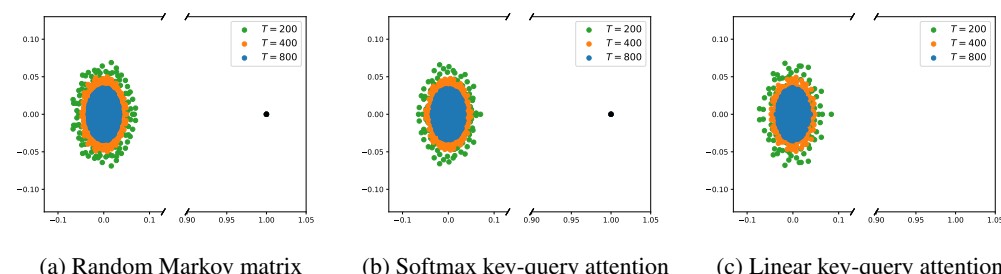

(a) Random Markov matrix     (b) Softmax key-query attention     (c) Linear key-query attention

Figure 1: As the size $T$ of a Random Markov matrix (Definition 1.1) grows, its eigenvalues form a circular bulk of radius $O(T^{-1/2})$ in the complex plane, except for the largest eigenvalue which remains 1 (the black dot in (a)). Proposition 1 demonstrates that applying the conventionalsoftmax key-query attention mechanism to orthonormal input tokens yields a Random Markov matrix exhibiting an outlier in its spectrum (b), in contrast to linear key-query attention (c). In practice, $T$ does not need to be too large for the limiting behaviour to appear, as shown above.

Note that not all random Markov matrix ensembles satisfy the conditions of Definition 1.1 which we colloquially refer to as Random Markov matrix. Remarkably, the standard key-query dot product attention matrix in the first layer of a transformer is a Random Markov matrix, as specified in the following.

**Proposition 1.** *Let* $\mathbf{X}_0 \in \mathbb{R}^{T \times d}$ *have orthonormal rows, i.e.* $\mathbf{X}_0 \mathbf{X}_0^\top = \mathbf{I}$. *Let*

$$\mathbf{A}_1(\mathbf{X}_0) := \mathrm{softmax}\left( \frac{\mathbf{X}_0 \mathbf{W}_1^Q \mathbf{W}_1^{K\top} \mathbf{X}_0^\top}{\sqrt{d_{qk}}} \right),$$

*where* $\mathbf{W}_1^Q, \mathbf{W}_1^K \in \mathbb{R}^{d \times d_{qk}}$ *have i.i.d.* $\mathcal{N}(0, \sigma_{qk}^2)$ *entries. Then* $\mathbf{A}_1(\mathbf{X}_0)$ *is a Random Markov matrix as in Definition 1.1 when* $d_{qk}$ *is large* [2]. *In particular, under Xavier or He scaling of* $\mathbf{W}_1^Q, \mathbf{W}_1^K$ *(Glorot & Bengio (2010); He et al. (2015), or any scaling such that* $\sigma_{qk}^2 \to 0$ *as* $d_{qk}$ *increases), the resulting* $\mathbf{A}_1(\mathbf{X}_0)$ *is degenerate, i.e. the "uniform attention"* $\frac{1}{T}\mathbf{1}_{T \times T}$.

The latter part of the above proposition was shown in (Noci et al., 2022, Lemma A.7) to justify their assumption of uniform attention, under which they demonstrate rank collapse in depth. In contrast, our analysis of the rank collapse is also valid for non-degenerate attention matrices. Based on the above proposition, much of our analysis is relevant to single-layer transformers using the standard key-query attention mechanism. However, as we will demonstrate in section 3, the case of multi-layer real-world transformers presents additional complexities, warranting further investigation.

It is shown in Bordenave et al. (2011) that the "bulk" of the limiting eigenvalue/singular value distribution of a Random Markov matrix matches (up to a scaling) that of an i.i.d. Gaussian matrix. Namely, if $\mathbf{A}$ is as in Definition 1.1 with variance $\sigma^2$, then $\sqrt{T}\mathbf{A}$ has the same bulk density as a Gaussian matrix with i.i.d. $\mathcal{N}(0, \frac{\sigma^2}{T})$ entries. Nonetheless, the "edge" of the spectrum of $\sqrt{T}\mathbf{A}$ behaves quite differently from the i.i.d. case. While the largest eigenvalue/singular value of an i.i.d. matrix is almost surely right at the boundary of its bulk, for Random Markov matrices there is a gap between the edge and the bulk given by Theorem 2. Without loss of generality, we formulate the theorem for Random Markov matrices with mean 1.

**Theorem 2** (Spectral gap in Random Markov matrices, Bordenave et al. (2011))**.** *Let* $\mathbf{A} \in \mathbb{R}^{T \times T}$ *be a Random Markov matrix whose underlying distribution has variance* $\sigma_A^2$. *Then,* $\lambda_1(\mathbf{A}) = 1$ *and almost surely,*

$$\lim_{T \to \infty} s_1(\mathbf{A}) = 1 \quad and \quad \lim_{T \to \infty} s_2(\sqrt{T}\mathbf{A}) = 2\sigma_A \quad while \quad \overline{\lim_{T \to \infty}} |\lambda_2(\sqrt{T}\mathbf{A})| \leq 2\sigma_A. \quad (2)$$

---

[2]Standard transformer implementations set $d_{qk} = d$, as detailed in the original paper by Vaswani et al. (2017). Therefore, assuming large $d_{qk}$ is not restrictive since we analyse the network in the large $d$ regime.

## 1.2 RELATED WORK

Rank collapse in transformers was first explored in Dong et al. (2021), where the authors show that the output of an attention-only transformer converges exponentially with depth to a single representation across tokens. A connection between rank collapse and vanishing gradients is made in Noci et al. (2022). By assuming uniform attention, the work of Noci et al. (2022) essentially reduces to proving rank collapse in depth, based on the assumption that rank collapse in width has already occurred (as opposed to showing why this premise holds). Our spectral analysis builds upon previous work that analysed the spectra of large random neural networks to better understand and stabilise initial training dynamics, such as Pennington et al. (2018) for fully-connected networks and Xiao et al. (2018) for convolutional networks.

It is also worth mentioning a line of research on possible alternatives to the softmax self-attention mechanism. Hron et al. (2020) speculate the advantage of ReLU and identity over softmax in training accuracy. He et al. (2022) proposes an ad hoc initialisation scheme tailored to prevent the token-wise covariance kernel from losing rank through layers. Besides, several practical works attempt to replace softmax-based key-query attention with faster options that surpass the so-called quadratic bottleneck, e.g. Peng et al. (2020); Choromanski et al. (2020); Katharopoulos et al. (2020). Furthermore, our provably effective adjustment to softmax by removing the spectral gap (or centering the output) has been independently suggested in Ali et al. (2023); Noci et al. (2024); Ye et al. (2024) as part of ad hoc solutions to stabilising signal propagation in transformers.

## 1.3 ORGANISATION OF THE PAPER

In section 2, we first introduce our model and reexamine the phenomena of rank collapse and exploding gradients, demonstrating that both occur with increasing depth in our transformer model at initialisation (Props. 3 and 4). Importantly, we diagnose for the first time an additional form of rank-collapse with increasing width, that we call rank collapse in width. We formulate its exact rate of decay in the context length as well as identify its root cause as being the spectral gap in softmax-based attention. Next, we introduce a modified attention mechanism that is specifically designed to have no spectral gap. We prove that this modification simultaneously resolves rank collapse in width, thus mitigating rank collapse in depth, and exploding gradients (Props. 5 and 6), thereby confirming the role of the spectral gap in such issues. Lastly, we study the spectra of the covariance kernel and the input-output Jacobian in our modified model (Props. 7 and 8) and discuss the possibility of further stabilising early training dynamics by tuning the initial distributions.

In section 3, we validate our findings, providing empirical evidence of rank collapse in both width and depth, as well as exploding gradients. We put to test our "remove the gap" solution across a range of architectures featuring LayerNorm and skip connections, discussing its possible training benefits. Finally, we present experiments that challenge the soundness of certain default scaling choices, such as Xavier initialization for the keys and queries, suggesting that they may require further revision in practice.

## 2 THEORETICAL RESULTS

We study as our model a deep attention-only single-head transformer at initialisation, where at each layer $\ell$, the signal is transformed as $\mathbf{X}_\ell = \mathbf{A}_\ell \mathbf{X}_{\ell-1} \mathbf{W}_\ell^V$. The input signal $\mathbf{X}_0 \in \mathbb{R}^{T \times d}$ has $T$ tokens of embedding dimension $d$, with a fixed ratio $\gamma := \frac{T}{d} \leq 1$. For a network of *depth* $L$, the input-output relationship is thus given by

$$\mathbf{X}_L = \mathbf{A}_L \mathbf{A}_{L-1} \dots \mathbf{A}_1 \mathbf{X}_0 \mathbf{W}_1^V \dots \mathbf{W}_{L-1}^V \mathbf{W}_L^V. \tag{3}$$

The *value matrices* $\mathbf{W}_\ell^V \in \mathbb{R}^{d \times d}$ are initialised independently with i.i.d. $\mathcal{N}(0, 1)$ entries and the *attention matrices* $\mathbf{A}_\ell \in \mathbb{R}^{T \times T}$ are independent Random Markov matrices with $\sigma_A^2 = 1$.

**Remark** (Scaling of value matrices). *The reason we initialise the value matrices with $\mathcal{N}(0, 1)$ entries rather than $\mathcal{N}(0, 1/d)$ (i.e. He initialisation) is that the attention matrices have singular values of magnitude $O(1/\sqrt{T})$ except for the leading one $s_1(\mathbf{A}) = 1 + o(1)$; see Theorem 2. So, in all but one direction, the attention matrix scales down the signal by a factor of $O(1/\sqrt{T})$, which will be compensated by $\mathbf{W}^V$ with singular values of magnitude $O(\sqrt{d})$.*

## 2.1 Revisiting Rank Collapse & Exploding Gradients

**Rank collapse.** As Theorem 2 reveals, for increasing $T$, a Random Markov matrix becomes effectively rank-one. We accurately describe the rate of this decay through the notion of *stable rank*, defined for any non-zero $\mathbf{M} \in \mathbb{R}^{m \times n}$ as

$$\operatorname{sr}(\mathbf{M}) := \frac{\|\mathbf{M}\|_F^2}{\|\mathbf{M}\|^2} = \frac{\sum_i s_i^2(\mathbf{M})}{s_1^2(\mathbf{M})}. \tag{4}$$

Naturally, any definition of "rank collapse" relies on a proxy for discrete rank, as the random matrices in question are almost surely full-rank, making it uninformative to refer to their actual rank. For example, Dong et al. (2021) consider the "one-infinity norm" of the residual (the difference between a matrix and its best approximation of the form $\mathbf{1}\mathbf{x}^\top$), defined as $\sqrt{\|\operatorname{res}(\mathbf{M})\|_1 \|\operatorname{res}(\mathbf{M})\|_\infty}$, while Noci et al. (2022) use $\sum_{i,j}(\mathbf{M}\mathbf{M}^\top)_{i,j}$, which is maximised when all rows of $\mathbf{M}$ are identical. We choose stable rank as our preferred proxy due to its clear geometrical interpretation and simple definition in terms of singular values.

Given an isometric input $\mathbf{X_0}$ with $\boldsymbol{\Sigma}_0 := \mathbf{X}_0\mathbf{X}_0^\top = \mathbf{I}$, we are interested in understanding how the stable rank of the covariance matrix at layer $\ell$,

$$\boldsymbol{\Sigma}_\ell := \mathbf{X}_\ell\mathbf{X}_\ell^\top,$$

evolves. Proposition 3 demonstrates how the stable rank collapses as the width $T$ increases.

**Proposition 3** (Rank collapse in width). *Assume $\boldsymbol{\Sigma}_0 = \mathbf{I}$. Then, for any $\ell \geq 1$,*

$$\lim_{T \to \infty} \operatorname{sr}(\boldsymbol{\Sigma}_\ell) = 1, \tag{5}$$

*with overwhelming probability*[3]. *Moreover, the convergence happens at a polynomial rate, i.e.* $\left|\operatorname{sr}(\boldsymbol{\Sigma}_\ell) - 1\right| = O(T^{1-4\ell})$.

**Exploding gradients.** A well-known issue that can disrupt training across various neural network architectures is the vanishing or exploding of gradients; see Hanin (2018). For attention-only transformers with degenerate attention, Noci et al. (2022) demonstrate that the gradients with respect to $\mathbf{W}_\ell^V$ vanish. Our model (3) allows for more general random attention while using a different scaling that makes the same quantity explode rather than vanish. Proposition 4 provides a lower bound on the rate at which the gradient grows.

**Proposition 4** (Exploding gradients). *For any $L \geq 2$ and $1 \leq \ell \leq L$, with overwhelming probability,*

$$\lim_{T \to \infty} \frac{1}{T^{L-1}} \left\| \frac{\partial \mathbf{X}_L}{\partial \mathbf{W}_\ell^V} \right\|_F^2 \geq C_{L-\ell}, \tag{6}$$

*for some constant $C_{L-\ell} > 0$. In particular, for $T$ large enough, $\|\partial \mathbf{X}_L/\partial \mathbf{W}_\ell^V\|_F^2$ diverges to infinity as $L$ increases. In the single-layer case $\ell = L = 1$, the following improved bound*

$$\lim_{T \to \infty} \frac{1}{T} \left\| \frac{\partial \mathbf{X}_1}{\partial \mathbf{W}_1^V} \right\|_F^2 \geq C \tag{7}$$

*holds almost surely.*

## 2.2 Attention Without the Gap

As previously seen, a Random Markov matrix $\mathbf{A} \in \mathbb{R}^{T \times T}$ can be written as

$$\mathbf{A} = \mathbb{E}\mathbf{A} + (\mathbf{A} - \mathbb{E}\mathbf{A}) = \frac{1}{T}\mathbf{1}_{T \times T} + \mathbf{A}^\perp, \tag{8}$$

where $\mathbf{1}_{T \times T} := (1, \cdots, 1)^\top(1, \cdots, 1)$ is the all-ones matrix and $\mathbf{A}^\perp$ has a limiting spectrum resembling that of a Gaussian matrix. Therefore, $\mathbf{A}$ is a rank-one perturbation of $\mathbf{A}^\perp$, whose spectral

---

[3]An event $E_n$ holds with overwhelming probability if, for every $A > 0$, $\mathbb{P}(E_n) \geq 1 - C_A n^{-A}$, for some constant $C_A$. As the name suggests, $E_n$ is more likely to hold if it occurs with *overwhelming* probability than with *high* probability, as defined in Tao (2012).

radius is $\lambda_1(\mathbf{A}^\perp) = O(T^{-1/2})$. Although the rank-one perturbation $\frac{1}{T}\mathbf{1}_{T\times T}$ cannot disturb the bulk of the spectrum, it causes the largest eigenvalue to "escape" from the bulk to 1, creating a spectral gap.

In light of this, we can slightly modify the attention mechanism to eliminate the outlier—and thus the gap—simply by replacing $\mathbf{A}$ with $\mathbf{A}^\perp$ at every layer, i.e.

$$\mathbf{X}_L^\perp = \mathbf{A}_L^\perp \mathbf{A}_{L-1}^\perp \ldots \mathbf{A}_1^\perp \mathbf{X}_0 \mathbf{W}_1^V \ldots \mathbf{W}_{L-1}^V \mathbf{W}_L^V, \tag{9}$$

where $\mathbf{A}_\ell^\perp := \mathbf{A}_\ell - T^{-1}\mathbf{1}_{T\times T}$. Note that this modification is applied only to the attention matrices (and *not* to the signal representation) and $\mathbf{X}_\ell^\perp$ serves as shorthand for the signal at layer $\ell \geq 1$ in a network whose $\mathbf{A}_\ell$'s are replaced with $\mathbf{A}_\ell^\perp$'s as in equation 9. We set $\mathbf{X}_0^\perp = \mathbf{X}_0$.

Since the modified attention exhibits no spectral gap (see Lemma 3 in section A.2), the stable rank of the covariance matrix $\mathbf{\Sigma}_\ell^\perp := \mathbf{X}_\ell^\perp \mathbf{X}_\ell^{\perp\top}$ no longer collapses to 1 in width, as detailed in Proposition 5 (cf. Proposition 3).

**Proposition 5** (Resolved rank collapse in width). *Let $\mathbf{X}_\ell^\perp = \mathbf{A}_\ell^\perp \mathbf{X}_{\ell-1}^\perp \mathbf{W}_\ell^V$ be the signal at layer $\ell$ in our modified model (9) and $\mathbf{\Sigma}_\ell^\perp := \mathbf{X}_\ell^\perp \mathbf{X}_\ell^{\perp\top} \in \mathbb{R}^{T\times T}$ be its covariance matrix. Then, almost surely, the rank does not collapse, i.e., there exists a constant $C_\ell > 0$ such that,*

$$\lim_{T\to\infty} \frac{\mathrm{sr}(\mathbf{\Sigma}_\ell^\perp)}{T} = C_\ell. \tag{10}$$

Our modification also mitigates the average growth of the gradients. Proposition 6 establishes a linear growth rate for $\|\partial\mathbf{X}_L^\perp / \partial\mathbf{W}_\ell^V\|_F^2$ in expectation, which should be compared to the rate of $T^{L-1}$ from Proposition 4.

**Proposition 6** (Resolved exploding gradients). *Let $\mathbf{X}_\ell^\perp = \mathbf{A}_\ell^\perp \mathbf{X}_{\ell-1}^\perp \mathbf{W}_\ell^V$ be the signal at layer $\ell$ in our modified model (9). Then, in expectation, the squared norm of the gradients grow linearly with $d$, i.e. there exists a constant $C > 0$ such that,*

$$\lim_{d\to\infty} \frac{1}{d}\mathbb{E}\left\|\frac{\partial\mathbf{X}_L^\perp}{\partial\mathbf{W}_\ell^V}\right\|_F^2 = C. \tag{11}$$

## 2.3 CAN TRANSFORMERS ACHIEVE DYNAMICAL ISOMETRY?

So far, we have established (i) the existence of an outlier eigenvalue/singular value in the spectrum of softmax-based attention matrices, and (ii) that removing this outlier helps with rank collapse and exploding gradients. In the absence of the outlier, we can take a further step to analyse the bulk of the spectra of the network's token-wise covariance and input-output Jacobian.

Let us assume that the input tokens are orthonormal, i.e. $\mathbf{\Sigma}_0 = \mathbf{X}_0\mathbf{X}_0^\top = \mathbf{I}$. As a criterion for faithful signal propagation, one should require that $\mathbf{\Sigma}_\ell^\perp$ stay close to the identity matrix. Considering the spectrum, this means that the limiting singular value distribution of $\mathbf{\Sigma}_\ell^\perp$ should concentrate around the value 1. A natural approach, as demonstrated in the fully-connected case in Pennington et al. (2017; 2018); Murray et al. (2022), is to adjust the model's hyperparameters to ensure that the mean of the limiting distribution is $O(1)$ and the variance is minimised. Proposition 7 describes the moments of the limiting singular value distribution of $\mathbf{\Sigma}_\ell^\perp$.

**Proposition 7** (Bulk of covariance kernel's singular value distribution). *Let $\mathbf{X}_\ell^\perp = \mathbf{A}_\ell^\perp \mathbf{X}_{\ell-1}^\perp \mathbf{W}_\ell^V$ be the signal at layer $\ell$ in our modified model (9) and $\mathbf{\Sigma}_\ell^\perp = \mathbf{X}_\ell^\perp \mathbf{X}_\ell^{\perp\top} \in \mathbb{R}^{T\times T}$ be its covariance matrix. Let the underlying Random Markov matrices $\mathbf{A}_\ell$ have variance $\sigma_A^2$ and $\mathbf{W}_\ell^V$ have i.i.d. $\mathcal{N}(0, \sigma_V^2)$ entries. Let $\mathbf{\Sigma}_0^\perp = \mathbf{I}$ and $\mathcal{D}_\ell$ be the limiting singular value distribution of $\mathbf{\Sigma}_\ell^\perp$. Then the mean and variance of $Z \sim \mathcal{D}_\ell$ are given by*

$$\mathbb{E}(Z) = (\sigma_A\sigma_V/\sqrt{\gamma})^{2\ell}, \tag{12}$$

$$\mathrm{Var}(Z) = \ell(1+\gamma)(\sigma_A\sigma_V/\sqrt{\gamma})^{4\ell}, \tag{13}$$

*where $\gamma := \frac{T}{d} \in (0, 1]$.*

The assumption $\gamma \leq 1$ is not essential and is made only to ensure that $\boldsymbol{\Sigma}_\ell^\perp$ is full-rank, avoiding trivial zero singular values. If $\gamma > 1$, then the limiting singular value distribution is given by $(1 - \gamma^{-1})\delta_0 + \gamma^{-1}\mathcal{D}_\ell$ and the mean and variance should be adjusted accordingly.

It is evident from the above proposition that simultaneously controlling both the mean and variance of $\mathcal{D}_\ell$ is not feasible. Model (9) does not have enough hyperparameters to achieve this balance. Indeed, to prevent the mean from growing or shrinking exponentially with depth, the product $\sigma_A \sigma_V$ must equal $\sqrt{\gamma}$. However, this constraint leads to the variance increasing linearly with $\ell$.

The Jacobian of the input-to-output function $f : \mathbf{X}_0 \mapsto \mathbf{X}_L^\perp$, represented by our modified transformer model, characterises the network's sensitivity to input perturbations up to first order, according to

$$f(\mathbf{X}_0 + \epsilon\mathbf{U}) \approx f(\mathbf{X}_0) + \epsilon\frac{\partial f}{\partial \mathbf{X}}\Big|_{\mathbf{X}_0}\mathbf{U}. \tag{14}$$

Let us consider the matricised version of the Jacobian at layer $\ell$, i.e.

$$\mathbf{J}_\ell := \frac{\partial \operatorname{vec}(\mathbf{X}_\ell^\perp)}{\partial \operatorname{vec}(\mathbf{X}_0)} \in \mathbb{R}^{Td \times Td}. \tag{15}$$

The goal is to ensure that the spectral energy of the Jacobian concentrates around 1, thereby minimising distortion of the input space geometry—a property often referred to as the *dynamical isometry* in the literature (see Pennington et al. (2017)). For our model (9), it is straightforward to show

$$\mathbf{J}_\ell = (\mathbf{A}_\ell^\perp \cdots \mathbf{A}_1^\perp) \otimes (\mathbf{W}_1^V \cdots \mathbf{W}_\ell^V) \in \mathbb{R}^{Td \times Td}, \tag{16}$$

where $\otimes$ denotes the Kronecker product. Proposition 8 describes the moments of the limiting squared singular value distribution of $\mathbf{J}_\ell$.

**Proposition 8** (Bulk of Jacobian's squared singular value distribution). *Let $\mathbf{X}_\ell^\perp = \mathbf{A}_\ell^\perp \mathbf{X}_{\ell-1}^\perp \mathbf{W}_\ell^V$ be the signal at layer $\ell$ in our modified model (9). Let the underlying Random Markov matrices $\mathbf{A}_\ell$ have variance $\sigma_A^2$ and $\mathbf{W}_\ell^V$ have i.i.d. $\mathcal{N}(0, \sigma_V^2)$ entries. Let $\mathcal{D}_\ell$ be the limiting distribution of the squared singular values of $\mathbf{J}_\ell := \partial \mathbf{X}_\ell^\perp / \partial \mathbf{X}_0{}^4$. Then the mean and variance of $Z \sim \mathcal{D}_\ell$ are given by*

$$\mathbb{E}(Z) = (\sigma_A \sigma_V)^{2\ell}, \tag{17}$$

$$\operatorname{Var}(Z) = \ell(\ell + 2)(\sigma_A \sigma_V)^{4\ell}. \tag{18}$$

Controlling the mean leads to a quadratically growing variance, while minimising the variance is only achievable if $\sigma_A \sigma_V < 1$, which, in turn, causes the mean to vanish. Without considering a more complex model, no choice of $(\sigma_A, \sigma_V)$ can achieve our goal of dynamical isometry.

## 3 EXPERIMENTS AND FURTHER INSIGHTS

**Rank collapse.** We highlight the practical relevance of our analysis by showing rank collapse occurs both in width and depth for famous transformer models like BERT, see Figure 3. As an input signal propagates through a transformer, we can address both forms of rank collapse—across width and depth—by eliminating the spectral gap induced by the attention matrix at each layer. Figures 2 and 4 reinforce our findings, showing that our removing the gap consistently mitigates rank collapse even in multi-layer transformers that include additional components such as LayerNorm, skip connections, or both. It is crucial to understand that rank collapse in depth is an inherent consequence of successive matrix multiplications. Therefore, architectural modifications can only slow the collapse rather than completely prevent it. We demonstrate this by showing that rank collapse in depth persists even when the attention matrix is set to the identity matrix—an extreme case with the highest possible stable rank and no spectral gap. Another possible way to slow down rank collapse in depth (though not eliminate it) is to set the value matrices as orthogonal matrices.

**Exploding gradients.** After passing an isometric input through the network, we compute the gradient norm as defined in equation 6. While our theory establishes a lower bound on the gradient norm at layer 1 that scales linearly with width, Figure 5 confirms an overall linear growth, supporting the order-optimality of our result. This linear trend persists even in more general settings that

---

[4] In a minor abuse of notation, we may write $\partial \mathbf{X}_\ell^\perp / \partial \mathbf{X}_0$ as a shorthand for $\partial \operatorname{vec}(\mathbf{X}_\ell^\perp) / \partial \operatorname{vec}(\mathbf{X}_0)$.

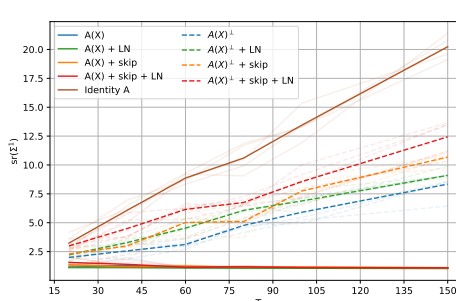
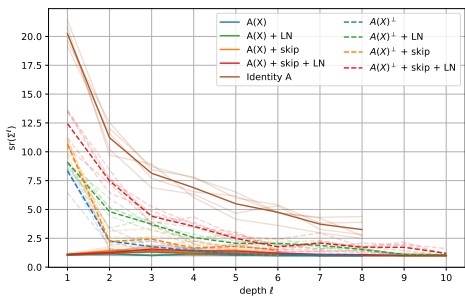

(a) Spectral gap implies rank collapse in width.     (b) The rank inevitably collapses in depth.

Figure 2: Rank collapse occurs both in width and depth. At layer one, our fix effectively prevents the rank from collapsing in width. Although rank collapse in depth occurs regardless of the presence of the spectral gap, our fix consistently slows the collapse—a feat no other module achieves.

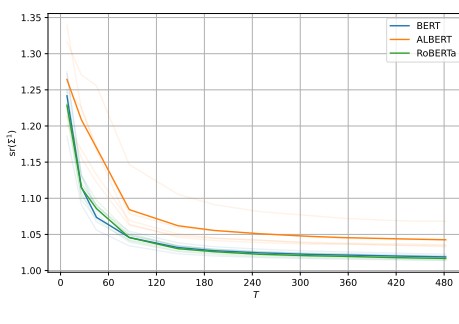
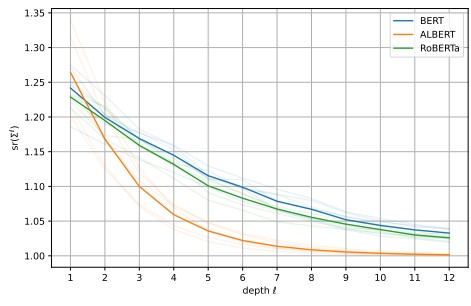

(a) Rank collapse in width.     (b) Rank collapse in depth.

Figure 3: Famous transformer encoders suffer from rank collapse at initialisation, both in (a) width and (b) depth. These untrained models are loaded from Hugging Face and sentences from this paper's abstract are tokenised using pre-trained tokenisers.

incorporate additional modules, such as LayerNorm or skip connections. Notably, the removal of the spectral gap affects only the slope of the gradient norm's growth, mirroring the behaviour observed in the lower bound derived in equations 4 and 6 for the single-layer case.

When the attention is a Random Markov matrix, gradient norms are effectively controlled with depth by either applying LayerNorm or removing the spectral gap, as illustrated in Figure 6, where the derived lower bound of $T^{L-1}$ is confirmed. Shifting to the more complex case of key-query attention in a multi-layer network, Figure 6 also demonstrates that our theoretical lower bound, derived for Random Markov matrices, remains valid. Interestingly, this bound becomes looser as depth increases, indicating that gradient norm explosion occurs at an even faster rate than predicted by our analysis; see the complementary Figure 10 in the Appendix. Moreover, in the multi-layer case, removing the largest singular value alone is no longer sufficient to prevent exploding gradients, suggesting more complex dynamics around the singular values of the attention matrix. One potential explanation is that the key-query attention spectrum now includes multiple outliers, whereas our method only addresses a single one. This hypothesis is explored further in the remainder of this section; see Figure 7.

**Training.** We evaluate our "remove the gap" proposal on a task designed to learn the entrywise Heaviside function; see section A.3 for implementation details. While our theoretical analysis does not address training dynamics, the experiments still offer valuable insights. In Figure 11, we present examples of training several architectural variants with and without our "remove the gap" solution. Further large-scale experiments are necessary to assess potential training benefits since the provided

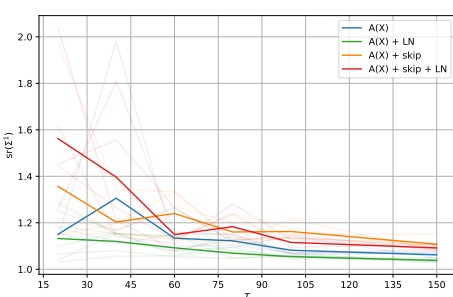

Figure 4: The rank collapses in width in the first layer across architectures.

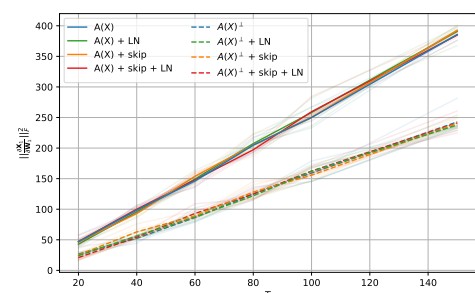

Figure 5: At layer one, the gradient norm scales linearly with width.

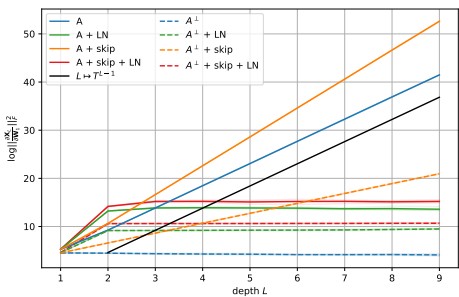

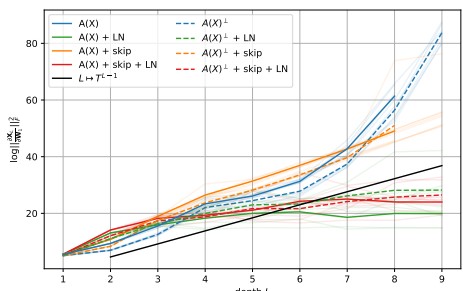

Figure 6: In multi-layer transformers with Random Markov attention (left), our "remove the gap" fix is effective, as we can precisely address the single outlier in the spectrum. However, with conventional key-query attention (right), the spectra of the attention matrices become more complex with depth, often exhibiting multiple outlier eigenvalues. This increased complexity reduces the effectiveness of our fix in controlling the gradient norm, as it only targets a single outlier, leaving other gaps untouched.

simulations are inconclusive. Beyond the first layer, key-query attention matrices are not Random Markov matrices, therefore their spectral properties are not well-known. For instance, we observe the emergence of additional outliers in the spectrum across layers, as shown in Figure 7. Investigating the configuration of the bulk and outliers in this context could lead to a natural solution for signal propagation issues by eliminating all outliers—an insight derived from our analysis.

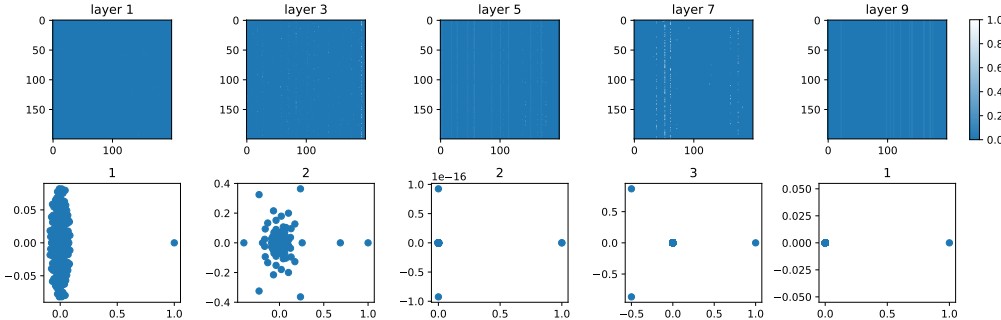

Figure 7: Entries (top row) of a $T \times T$ key-query attention matrix (with $T = 100$), along with its spectrum (bottom row) across layers. We indicate the number of eigenvalues whose magnitudes exceed a threshold of $0.5$ to signify the presence of multiple outliers. The layer-wise evolution of the spectrum requires further study, however, the matrix consistently tends to uniform attention for large $\ell$.

**Scaling discussion.**   Our analysis provides practitioners with valuable insights into the scaling of some key quantities in transformer architectures. First, in Figure 8a, we plot the training loss of a single-layer transformer with key-query attention, for which our theory effectively predicts signal propagation at the initialisation stage. Based on Proposition 1, when the keys $\mathbf{W}^K$ and queries $\mathbf{W}^Q$ are initialised using Xavier's initialisation scheme, the attention matrix rapidly converges to degeneracy. Consequently, removing the spectral gap in this case effectively reduces the attention matrix to 0. This explains the plateau in the training curve during the initial steps for the Xavier-initialised model combined with the spectral gap removal. On the other hand, our proposed fix achieves performance similar to that of the original variant. Another scaling that requires refinement is that of a skip connection. Traditionally, the value matrices are sampled from $\mathcal{N}(0, d^{-1})$, the attention matrix is softmax-based, and the signal propagates from one layer $\ell$ to the next as

$$\mathbf{X}_{\ell+1} = \mathbf{A}_{\ell+1}(\mathbf{X}_\ell)\mathbf{X}_\ell \mathbf{W}^V_{\ell+1} + \mathbf{X}_\ell.$$

In this scenario, starting from an isometric input $\mathbf{X}_0$, the attention mechanism is down-weighted by a factor of $\sqrt{d}$ relative to the skip branch, i.e. the signal coming from the previous layer. Therefore, if the attention mechanism is to be fairly represented in the signal's propagation through depth, the values should instead be drawn from $\mathcal{N}(0, 1)$. If the value weights are sampled from $\mathcal{N}(0, d^{-1})$, the skip connection becomes what we refer to as "upscaled", meaning each layer essentially passes information from the previous layer without significant transformation. This severely limits the model's expressivity, reducing its capacity to learn nonlinear mappings. In Figure 8b, we confirm this observation by comparing the training losses associated with each option. It is not surprising that the "upscaled skip" variants perform worse, as we are attempting to learn a nonlinear function using a virtually linear model.

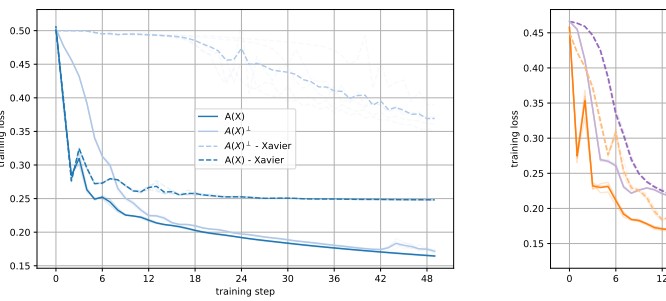 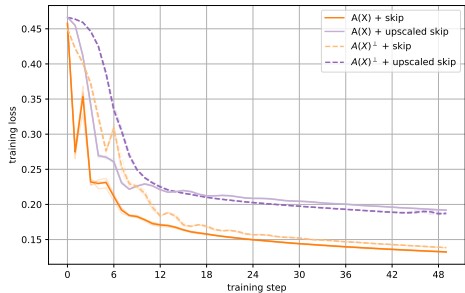

(a) Xavier initialisation over the keys and queries.       (b) Scaling of the skip connection.

Figure 8: Studying transformers through a spectral lens raises important questions about the soundness of some default scaling choices.

## 4    CONCLUSION

We introduced a new mathematical framework for studying the self-attention mechanism at initialisation, leveraging results from random matrix theory and free probability. By analysing the spectral properties of Random Markov matrices, we diagnosed random softmax-based attention with a spectral gap that leads to rank collapse in width—a phenomenon revealed and demonstrated for the first time by our analysis—alongside the previously established rank collapse in depth and exploding gradients.

We proposed a straightforward modification of the attention mechanism, which proved effective in slowing rank collapse when the spectrum contains a single outlier. Additionally, we observed that the spectra of standard key-query attention matrices often feature multiple outliers. Our experiments also pointed to potential issues with some common initialisation schemes for transformers. We hope our work encourages the community to adopt a spectral perspective in investigating more complex transformer architectures and attention models.

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

# A  APPENDIX

## A.1  PROOFS

*Proof of Proposition 1.* Let us show that the attention matrix $\mathbf{A}_1(\mathbf{X}_0)$ satisfies Definition 1.1 by demonstrating that the random variables

$$Z_{i,j} := \exp\left(\frac{\mathbf{X}_0\mathbf{W}_1^Q\mathbf{W}_1^{K\top}\mathbf{X}_0^\top}{\sqrt{d_{qk}}}\right)_{i,j}$$

are i.i.d. with a finite fourth moment.

Since the key and query matrices are initialised as Gaussian i.i.d. matrices and the input data $\mathbf{X}_0$ is isometric, $\widetilde{\mathbf{W}}^Q := \mathbf{X}_0\mathbf{W}_1^Q$ and $\widetilde{\mathbf{W}}^K := \mathbf{X}_0\mathbf{W}_1^K$ follow the same distribution as $\mathbf{W}_1^Q$ and $\mathbf{W}_1^K$. Each $Z_{i,j}$ can be written as the exponential of the inner product between the $i$-th row of $\widetilde{\mathbf{W}}^Q$ and the $j$-th row of $\widetilde{\mathbf{W}}^K$, thus $Z_{i,j}$ are i.i.d. and we only need to prove that $\mathbb{E}|Z_{1,1}|^4 < \infty$. Let us define

$$U_{d_{qk}} := \sum_{r=1}^{d_{qk}} \widetilde{\mathbf{W}}_{1,r}^Q \widetilde{\mathbf{W}}_{1,r}^K$$

to be the dot product of the first row of $\widetilde{\mathbf{W}}^Q$ and the first row of $\widetilde{\mathbf{W}}^K$. So, $U_{d_{qk}}$ is simply the sum of $d_{qk}$ i.i.d. copies of $U_1$, the product of two independent Gaussian random variables, whose density is known to be

$$f_1(x) := \frac{1}{\pi\sigma_{qk}^2} K_0\left(\frac{|x|}{\sigma_{qk}^2}\right),$$

where $K_0$ is the modified Bessel function of the second kind. Therefore, the probability density function of $U_{d_{qk}}$ is given by the $d_{qk}$-fold convolution

$$f_{d_{qk}}(x) = \underbrace{f_1(x) * \cdots * f_1(x)}_{d_{qk} \text{ times}}.$$

It is also known that $K_0(x)$ asymptotically behaves like $\sqrt{\frac{\pi}{2x}}e^{-x}$ and that the convolution $g * h$ decays at least as fast as the slower of $g$ and $h$. Combining these facts, we conclude that $f_{d_{qk}}$ decays at least as fast as $e^{-x}$, i.e.

$$f_{d_{qk}}(x) = g(|x|)e^{-|x|},$$

for some polynomially-bounded $g$. Now we can bound our quantity of interest

$$\mathbb{E}|Z_{1,1}|^4 = \mathbb{E}\left[\exp\left(\frac{4U_{d_{qk}}}{\sqrt{d_{qk}}}\right)\right]$$

$$= \int_{\mathbb{R}} e^{4x/\sqrt{d_{qk}}}\, g(|x|)e^{-|x|}dx$$

$$< \infty,$$

as long as $\frac{4}{\sqrt{d_{qk}}} < 1$, i.e. $d_{qk} > 16$. $\qquad\square$

*Proof of Proposition 3.* Fix $\ell \geq 1$. By definition of stable rank, we have

$$\text{sr}(\mathbf{\Sigma}_\ell) = \frac{\sum_{i=1}^T s_i^2(\mathbf{\Sigma}_\ell)}{s_1^2(\mathbf{\Sigma}_\ell)} = \frac{\sum_{i=1}^T s_i^2(\mathbf{X}_\ell\mathbf{X}_\ell^\top)}{s_1^2(\mathbf{X}_\ell\mathbf{X}_\ell^\top)} = \frac{\sum_{i=1}^T s_i^4(\mathbf{X}_\ell)}{s_1^4(\mathbf{X}_\ell)}$$

$$= 1 + \sum_{i=2}^T \frac{s_i^4(\mathbf{X}_\ell)}{s_1^4(\mathbf{X}_\ell)} \leq 1 + (T-1)\frac{s_2^4(\mathbf{X}_\ell)}{s_1^4(\mathbf{X}_\ell)}.$$

For $T$ large enough, let us say bigger than some $T_0$, Theorem 10 provides a deterministic upper bound, i.e. $s_2(\mathbf{X}_\ell) \leq K$ for some constant $K$. Moreover, Theorem 9 gives the bound $T^{-\ell}s_1(\mathbf{X}_\ell) \in$

$(1 - t, 1 + t)$ with (overwhelming) probability $P_{t,T}$ for arbitrary $t > 0$ and $T$ bigger than some $T_1$. Thus, for $T \geq \max(T_0, T_1)$,

$$1 \leq \mathrm{sr}(\mathbf{\Sigma}_\ell) \leq 1 + \frac{(T-1)K^4}{T^{4\ell}(1-t)^4}$$

with probability at least $P_{t,T}$. Therefore, the event

$$\lim_{T \to \infty} \mathrm{sr}(\mathbf{\Sigma}_\ell) = 1$$

holds with overwhelming probability. □

*Proof of Proposition 4.* Note that we will treat the matrix-to-matrix derivatives such as $\partial \mathbf{X}_L / \partial \mathbf{W}_\ell^V$ not as a tensor (in $\mathbb{R}^{T \times d \times d \times d}$), but as its matricised version (in $\mathbb{R}^{Td \times d^2}$). We make use of the chain rule to compute the gradients of interest. Namely, at layer $\ell$,

$$\frac{\partial \mathbf{X}_L}{\partial \mathbf{W}_\ell^V} = \frac{\partial \mathbf{X}_L}{\partial \mathbf{X}_\ell} \frac{\partial \mathbf{X}_\ell}{\partial \mathbf{W}_\ell^V}$$

$$= \left( \left( \mathbf{A}_L \dots \mathbf{A}_{\ell+1} \right) \otimes \left( \mathbf{W}_{\ell+1}^V \dots \mathbf{W}_L^V \right) \right) \left( \left( \mathbf{A}_\ell \dots \mathbf{A}_1 \mathbf{X}_0 \mathbf{W}_1^V \dots \mathbf{W}_{\ell-1}^V \right) \otimes \mathbf{I}_d \right)$$

$$= \big( \underbrace{\mathbf{A}_L \dots \mathbf{A}_1 \mathbf{X}_0 \mathbf{W}_1^V \dots \mathbf{W}_{\ell-1}^V}_{:=\mathbf{P}_1} \big) \otimes \big( \underbrace{\mathbf{W}_{\ell+1}^V \dots \mathbf{W}_L^V}_{:=\mathbf{P}_2} \big).$$

Then, by properties of Kronecker product, we have

$$\left\| \frac{\partial \mathbf{X}_L}{\partial \mathbf{W}_\ell^V} \right\|_F^2 = \sum_i s_i^2 \left( \frac{\partial \mathbf{X}_L}{\partial \mathbf{W}_\ell^V} \right) = \sum_{i,j} s_i^2(\mathbf{P}_1) s_j^2(\mathbf{P}_2) \geq s_1^2(\mathbf{P}_1) s_1^2(\mathbf{P}_2). \quad (19)$$

The largest singular value of a product of i.i.d. Gaussian matrices has been studied extensively, e.g., see Akemann et al. (2013); Młotkowski et al. (2015). Nait Saada & Naderi (2024) show that, almost surely,

$$s_1^2(\mathbf{P}_2) = T^{L-\ell} \frac{(L-\ell+1)^{L-\ell+1}}{(L-\ell)^{L-\ell}} (1 + o(1)).$$

On the other hand, by Theorem 9, $s_1(\mathbf{P}_1)$ concentrates around $T^{\frac{\ell-1}{2}}$ with overwhelming probability, i.e., for $T$ large enough,

$$s_1^2(\mathbf{P}_1) \in (T^{\ell-1}(1-t)^2, T^{\ell-1}(1+t)^2),$$

with probability at least $P_{t,T}$. Altogether, with an overwhelming probability we have

$$\lim_{T \to \infty} \frac{1}{T^{L-1}} \left\| \frac{\partial \mathbf{X}_L}{\partial \mathbf{W}_\ell^V} \right\|_F^2 \geq (1-t)^2 \frac{(L-\ell+1)^{L-\ell+1}}{(L-\ell)^{L-\ell}}.$$

One can get a better bound in the single-layer case ($\ell = L = 1$). Since $\mathbf{P}_2 = \mathbf{I}_d$, we can rewrite equation 19 as

$$\left\| \frac{\partial \mathbf{X}_1}{\partial \mathbf{W}_1^V} \right\|_F^2 = \sum_i s_i^2 \left( \frac{\partial \mathbf{X}_1}{\partial \mathbf{W}_1^V} \right) = \sum_i s_i^2(\mathbf{P}_1) \sum_j s_j^2(\mathbf{I}_d) \geq d \cdot s_1^2(\mathbf{P}_1),$$

while $s_1^2(\mathbf{P}_1) = s_1^2\left( \mathbf{A}_1(\mathbf{X}_0)\mathbf{X}_0 \right) = O(1)$ almost surely. Therefore, almost surely, the following improved bound

$$\lim_{T \to \infty} \frac{1}{T} \left\| \frac{\partial \mathbf{X}_1}{\partial \mathbf{W}_1^V} \right\|_F^2 \geq C,$$

holds. □

*Proof of Proposition 5.* The resolved stable rank can be written as,

$$\frac{\mathrm{sr}(\mathbf{\Sigma}_\ell^\perp)}{T} = \frac{T^{-1} \sum_{i=1}^T s_i^2(\mathbf{A}_\ell^\perp \dots \mathbf{A}_1^\perp \mathbf{X}_0 \mathbf{W}_1^V \dots \mathbf{W}_\ell^V)}{s_1^2(\mathbf{A}_\ell^\perp \dots \mathbf{A}_1^\perp \mathbf{X}_0 \mathbf{W}_1^V \dots \mathbf{W}_\ell^V)}.$$

By submultiplicativity of the operator norm,

$$\frac{\mathrm{sr}(\boldsymbol{\Sigma}_\ell^\perp)}{T} \geq \frac{T^{-1}\sum_{i=1}^T s_i^2(\mathbf{A}_\ell^\perp \ldots \mathbf{A}_1^\perp \mathbf{X}_0 \mathbf{W}_1^V \ldots \mathbf{W}_\ell^V)}{s_1^2(\mathbf{A}_\ell^\perp)\ldots s_1^2(\mathbf{A}_1^\perp)s_1^2(\mathbf{X}_0\mathbf{W}_1^V)\ldots s_1^2(\mathbf{W}_\ell^V)}.$$

Let us call $P_T$ the fraction of squared singular values of $\mathbf{A}_\ell^\perp \ldots \mathbf{A}_1^\perp \mathbf{X}_0 \mathbf{W}_1^V \ldots \mathbf{W}_\ell^V$ above a certain finite threshold $c$, i.e.

$$P_T := \frac{1}{T}\sum_{i=1}^T \mathbb{1}_{s_i^2(\mathbf{A}_\ell^\perp \ldots \mathbf{A}_1^\perp \mathbf{X}_0 \mathbf{W}_1^V \ldots \mathbf{W}_\ell^V) > c}.$$

Then, trivially

$$\frac{\mathrm{sr}(\boldsymbol{\Sigma}_\ell^\perp)}{T} \geq \frac{c\, P_T}{s_1^2(\mathbf{A}_\ell^\perp)\ldots s_1^2(\mathbf{A}_1^\perp)s_1^2(\mathbf{X}_0\mathbf{W}_1^V)\ldots s_1^2(\mathbf{W}_\ell^V)}. \tag{20}$$

Assuming the asymptotic freeness of all attention matrices $\mathbf{A}_1^\perp, \ldots, \mathbf{A}_\ell^\perp$ and weight matrices $\widetilde{\mathbf{W}}_1^V = \mathbf{X}_0\mathbf{W}_1, \mathbf{W}_2, \ldots, \mathbf{W}_\ell$, we may write the limiting squared singular value distribution of $\mathbf{X}_\ell^\perp$ as the free convolution of the corresponding Marchenko-Pastur distributions:

$$\mathcal{M} := \mathcal{MP}^{\boxtimes \ell}(1, \sigma_A) \boxtimes \mathcal{MP}(\gamma, \frac{\sigma_W}{\sqrt{\gamma}}) \boxtimes \mathcal{MP}^{\boxtimes \ell-1}(1, \frac{\sigma_W}{\sqrt{\gamma}}).$$

Then, almost surely,

$$P_T \longrightarrow P := \int_c^\infty d\mathcal{M}.$$

The distribution $\mathcal{M}$ is compactly supported on the interval $[0, s_{\gamma,2\ell}^+]$, where $s_{\gamma,2\ell}^+$ does not depend on $T$. So, by choosing $c < s_{\gamma,2\ell}^+$, we can make $c\,P$ a non-zero constant. Moreover, the denominator of equation 20 converges almost surely to some constant (in $T$), i.e.

$$s_1^2(\mathbf{A}_\ell^\perp)\ldots s_1^2(\mathbf{A}_1^\perp)s_1^2(\mathbf{X}_0\mathbf{W}_1^V)\ldots s_1^2(\mathbf{W}_\ell^V) \to (2\sigma_A)^{2\ell}\sigma_W^{2\ell}2^{2(\ell-1)}(1+\gamma^{-1/2})^2.$$

Thus, almost surely,

$$\lim_{T\to\infty}\frac{\mathrm{sr}(\boldsymbol{\Sigma}_\ell^\perp)}{T} \geq \frac{c\,P}{(\sigma_A\sigma_W)^{2\ell}4^{\ell-1}(1+\gamma^{-1/2})^2} > 0.$$

$\square$

*Proof of Proposition 6.* Let us compute the resolved gradients:

$$\frac{\partial \mathbf{X}_L^\perp}{\partial \mathbf{W}_\ell^V} = \frac{\partial \mathbf{X}_L^\perp}{\partial \mathbf{X}_\ell^\perp}\frac{\partial \mathbf{X}_\ell^\perp}{\partial \mathbf{W}_\ell^V}$$

$$= \big(\underbrace{\mathbf{A}_L^\perp \ldots \mathbf{A}_1^\perp \mathbf{X}_0 \mathbf{W}_1^V \ldots \mathbf{W}_{\ell-1}^V}_{:=\mathbf{P}_1^\perp}\big) \otimes \big(\underbrace{\mathbf{W}_{\ell+1}^V \ldots \mathbf{W}_L^V}_{=\mathbf{P}_2}\big).$$

Therefore,

$$\mathbb{E}\left\|\frac{\partial \mathbf{X}_L^\perp}{\partial \mathbf{W}_\ell^V}\right\|_F^2 = \mathbb{E}\Big[\mathrm{tr}\Big(\frac{\partial \mathbf{X}_L^\perp}{\partial \mathbf{W}_\ell^V}\big(\frac{\partial \mathbf{X}_L^\perp}{\partial \mathbf{W}_\ell^V}\big)^\top\Big)\Big]$$

$$= \mathbb{E}\Big[\mathrm{tr}\big(\mathbf{P}_1^\perp(\mathbf{P}_1^\perp)^\top\big)\mathrm{tr}\big(\mathbf{P}_2\mathbf{P}_2^\top\big)\Big].$$

Assuming $\mathbf{P}_1^\perp$ and $\mathbf{P}_2$ are asymptotically free, we have

$$\lim_{d\to\infty}\frac{1}{d^2}\mathbb{E}\left\|\frac{\partial \mathbf{X}_L^\perp}{\partial \mathbf{W}_\ell^V}\right\|_F^2 = \lim_{d\to\infty}\frac{1}{d}\mathbb{E}\Big(\mathrm{tr}\big(\mathbf{P}_1^\perp(\mathbf{P}_1^\perp)^\top\big)\Big)\lim_{d\to\infty}\frac{1}{d}\mathbb{E}\Big(\mathrm{tr}\big(\mathbf{P}_2\mathbf{P}_2^\top\big)\Big).$$

For each product matrix, the normalised expectation on the RHS of the above converges to the first moment of its limiting squared singular value distribution. By scaling them properly, i.e.

$$\widetilde{\mathbf{P}}_1^\perp := \sqrt{T}\mathbf{A}_L^\perp \ldots \sqrt{T}\mathbf{A}_1^\perp \frac{1}{\sqrt{d}}\mathbf{X}_0\mathbf{W}_1^V \ldots \frac{1}{\sqrt{d}}\mathbf{W}_{\ell-1}^V = T^{L/2}d^{-(\ell-1)/2}\mathbf{P}_1^\perp,$$

$$\widetilde{\mathbf{P}}_2 := \frac{1}{\sqrt{d}}\mathbf{W}_{\ell+1}^V \ldots \frac{1}{\sqrt{d}}\mathbf{W}_L^V = d^{-(L-\ell)/2}\mathbf{P}_2,$$

we make sure that those limiting distributions (free convolutions of Marchenko-Pastur distributions) are compactly supported on an interval of length $O(1)$ and, hence, both $C_1 := \lim \mathbb{E}(\mathrm{tr}(\widetilde{\mathbf{P}}_1^{\perp}(\widetilde{\mathbf{P}}_1^{\perp})^{\top}))$ and $C_2 := \lim \mathbb{E}(\mathrm{tr}(\widetilde{\mathbf{P}}_2(\widetilde{\mathbf{P}}_2)^{\top}))$ are constants. Thus, since $T = \gamma d$,

$$\lim_{d \to \infty} \frac{1}{d^2} \mathbb{E} \left\| \frac{\partial \mathbf{X}_L^{\perp}}{\partial \mathbf{W}_\ell^V} \right\|_F^2 = C_1(T^{-L}d^{\ell-1}) \cdot C_2(d^{L-\ell}) = Cd^{-1},$$

or

$$\lim_{d \to \infty} \frac{1}{d} \mathbb{E} \left\| \frac{\partial \mathbf{X}_L^{\perp}}{\partial \mathbf{W}_\ell^V} \right\|_F^2 = C.$$

$\square$

*Proof of Theorem 7.* Since $T = \gamma d$, we can write

$$\mathbf{X}_\ell = \mathbf{A}_\ell^{\perp} \dots \mathbf{A}_1^{\perp} \mathbf{X}_0 \mathbf{W}_1^V \dots \mathbf{W}_\ell^V$$
$$= \sqrt{T} \mathbf{A}_\ell^{\perp} \dots \sqrt{T} \mathbf{A}_1^{\perp} \frac{1}{\sqrt{d}} \left( \frac{1}{\sqrt{\gamma}} \mathbf{X}_0 \mathbf{W}_1^V \right) \dots \frac{1}{\sqrt{d}} \left( \frac{1}{\sqrt{\gamma}} \mathbf{W}_\ell^V \right).$$

Each of the rescaled matrices above has squared singular values that almost surely follow a Marchenko-Pastur distribution $\mathcal{MP}(p, \alpha)$, where $p$ is the ratio between the numbers of rows and columns of each matrix, and $\alpha$ the variance of its entries. Therefore, almost surely, the squared singular values of $\mathbf{X}_\ell$, or equivalently the singular values of $\boldsymbol{\Sigma}_\ell$, follow a distribution $\mathcal{M}$ which is given by the free convolution

$$\mathcal{M} := \mathcal{MP}^{\boxtimes \ell}(1, \sigma_A) \boxtimes \mathcal{MP}(\gamma, \sigma_V/\sqrt{\gamma}) \boxtimes \mathcal{MP}^{\boxtimes \ell-1}(1, \sigma_V/\sqrt{\gamma}).$$

The moments of such a distribution are given by Lemma 4 in the general case. Substituting the corresponding values from our setting gives the desired result.

$\square$

*Proof of Theorem 8.* Let $\mathbf{A}^{\perp} := \mathbf{A}_\ell^{\perp} \dots \mathbf{A}_1^{\perp} \in \mathbb{R}^{T \times T}$ and $\mathbf{W}^V := \mathbf{W}_1^V \dots \mathbf{W}_\ell^V \in \mathbb{R}^{d \times d}$. Then

$$\mathbf{J}_\ell = \mathbf{A}^{\perp} \otimes \mathbf{W}^V \in \mathbb{R}^{Td \times Td},$$

and we can compute the $k$-th moment of its limiting squared singular value distribution as

$$\lim_{T,d \to \infty} \mathbb{E}\left[ \frac{1}{Td} \mathrm{tr}(\mathbf{J}_\ell \mathbf{J}_\ell^{\top})^k \right] = \lim_{T,d \to \infty} \mathbb{E}\left[ \frac{1}{Td} \mathrm{tr}\left( (\mathbf{A}^{\perp} \mathbf{A}^{\perp \top} \otimes \mathbf{W}^V \mathbf{W}^{V \top})^k \right) \right]$$
$$= \lim_{T,d \to \infty} \mathbb{E}\left[ \frac{1}{T} \mathrm{tr}\left( (\mathbf{A}^{\perp} \mathbf{A}^{\perp \top})^k \right) \frac{1}{d} \mathrm{tr}\left( (\mathbf{W}^V \mathbf{W}^{V \top})^k \right) \right],$$

using simple linear algebra. Under the assumption that the matrices $\mathbf{A}$ and $\mathbf{W}^V$ are asymptotically free, the above limiting moment can be written as the product of individual limiting moments, i.e.

$$\lim_{T,d \to \infty} \mathbb{E}\left[ \frac{1}{Td} \mathrm{tr}(\mathbf{J}_\ell \mathbf{J}_\ell^{\top})^k \right] = \lim_{T \to \infty} \mathbb{E}\left[ \frac{1}{T} \mathrm{tr}\left( (\mathbf{A}^{\perp} \mathbf{A}^{\perp \top})^k \right) \right] \lim_{d \to \infty} \mathbb{E}\left[ \frac{1}{d} \mathrm{tr}\left( (\mathbf{W}^V \mathbf{W}^{V \top})^k \right) \right],$$

where each factor equals the $k$-th moment of the limiting squared singular value distribution of its respective matrix. For both $\mathbf{A}^{\perp}$ and $\mathbf{W}^V$ the limits exist almost surely, and are equal (up to a variance factor) to the well-known Fuss-Catalan numbers, defined by

$$\mathrm{FC}_\ell(k) := \frac{1}{\ell k + 1} \binom{\ell k + k}{k}.$$

Therefore, almost surely,

$$\lim_{T,d \to \infty} \mathbb{E}\left[ \frac{1}{Td} \mathrm{tr}(\mathbf{J}_\ell \mathbf{J}_\ell^{\top})^k \right] = (\sigma_A^2)^k \mathrm{FC}_\ell(k) \times (\sigma_V^2)^k \mathrm{FC}_\ell(k).$$

Simple calculations in the case $k = 1$ and $k = 2$ yield the specified formulae for mean and variance.

$\square$

## A.2 LEMMAS

**Lemma 1.** *Let $\mathbf{W}_1 \in \mathbb{R}^{T \times d}$ and $\mathbf{W}_2, \ldots, \mathbf{W}_q \in \mathbb{R}^{d \times d}$ be independent Gaussian matrices with i.i.d. $\mathcal{N}(0, 1)$ entries, and $\mathbf{u} \in \mathbb{R}^T$ a unit vector. Then,*

$$\mathbb{E}\big[s_1^2(\mathbf{u}\mathbf{u}^\top \mathbf{W}_1 \ldots \mathbf{W}_q)\big] = d^q, \tag{21}$$

*and the event*

$$\left|\frac{s_1(\mathbf{u}\mathbf{u}^\top \mathbf{W}_1 \ldots \mathbf{W}_q)}{d^{q/2}} - 1\right| < t$$

*holds with overwhelming probability.*

*Proof.* First of all, note that the distribution of $s_1(\mathbf{u}\mathbf{u}^\top \mathbf{W}_1 \ldots \mathbf{W}_q)$ is independent of the choice of $\mathbf{u}$, since $\mathbf{W}_1, \ldots, \mathbf{W}_q$ are rotation-invariant. Let us write $\mathbf{u}^\top \mathbf{W}_1 = \alpha_1 \mathbf{u}_1^\top$, where $\mathbf{u}_1 \in \mathbb{R}^d$ has length 1. Similarly, define

$$\alpha_{i+1} := \|\mathbf{u}_i^\top \mathbf{W}_{i+1}\|_2, \quad \mathbf{u}_{i+1}^\top := \frac{\mathbf{u}_i^\top \mathbf{W}_{i+1}}{\alpha_{i+1}},$$

for $1 \le i \le q-1$. So, we can write

$$\begin{aligned}
s_1(\mathbf{u}\mathbf{u}^\top \mathbf{W}_1 \mathbf{W}_2 \ldots \mathbf{W}_q) &= s_1(\mathbf{u}(\alpha_1 \mathbf{u}_1^\top) \mathbf{W}_2 \ldots \mathbf{W}_q) \\
&= s_1(\mathbf{u}(\alpha_1 \alpha_2 \mathbf{u}_2^\top)) \ldots \mathbf{W}_q) \\
&= \ldots \\
&= \alpha_1 \ldots \alpha_q \cdot s_1(\mathbf{u}\mathbf{u}_q^\top) \\
&= \alpha_1 \ldots \alpha_q,
\end{aligned}$$

where $s_1(\mathbf{u}\mathbf{u}_q^\top) = 1$ since $\mathbf{u}\mathbf{u}_q^\top$ naturally takes the form of an SVD with a single nonzero singular value equal to 1. The random variables $\alpha_1, \ldots, \alpha_q$ are independent (by independence of $\mathbf{W}_i$'s) and identically distributed (by rotation-invariance of $\mathbf{W}_i$'s). Without loss of generality, we can substitute $\mathbf{e}_1$ (the first column of the identity matrix) for $\mathbf{u}$ or $\mathbf{u}_i$ to get

$$\alpha_i \overset{d}{=} \|\mathbf{e}_1^\top \mathbf{W}_i\| = \|\mathbf{w}\|$$

where $\mathbf{w} \in \mathbb{R}^d$ (the first row of $\mathbf{W}_i$) has i.i.d. $\mathcal{N}(0, 1)$ entries. Thus, $\mathbb{E}(\alpha_i^2) = \mathbb{E}(\|\mathbf{w}\|_2^2) = d$, and by independence of $\alpha_i$'s we have

$$\mathbb{E}\big[s_1^2(\mathbf{u}\mathbf{u}^\top \mathbf{W}_1 \ldots \mathbf{W}_q)\big] = d^q.$$

Moreover, since each $\alpha_i^2$ has a chi-squared distribution with $d$ degrees of freedom, we can write it as the sum of $d$ independent squared standard Gaussian random variables $\alpha_i^2 = \sum_{j=1}^d w_{i,j}^2$. Thus,

$$s_1^2(\mathbf{u}\mathbf{u}^\top \mathbf{W}_1 \ldots \mathbf{W}_q) = \prod_{i=1}^q \alpha_i^2 = \prod_{i=1}^q (w_{i,1}^2 + \cdots + w_{i,d}^2) = \sum_{j=1}^{d^q} Z_j^2,$$

where each $Z_j$ is the product of $q$ independent $\mathcal{N}(0, 1)$ random variables, and therefore is sub-Weibull with parameter $2/q$. We shall apply generalised Bernstein's inequality for the normalised sum of mean-zero sub-Weibull random variables Kuchibhotla & Chakrabortty (2022); Bong & Kuchibhotla (2023), i.e.

$$\mathbb{P}\Big(\Big|\frac{1}{N}\sum_{i=1}^N X_i\Big| \ge u\Big) \le 2\exp\big[-CN\min(\frac{u^2}{K^2}, \frac{u^\beta}{K^\beta})\big], \tag{22}$$

where $X_i$'s are independent mean-zero sub-Weibull random variables with parameter $\beta$ and $K := \max_i \|X_i\|_{\psi_\beta}$. Applying equation 22 on

$$\frac{1}{d^q} s_1^2(\mathbf{u}\mathbf{u}^\top \mathbf{W}_1 \ldots \mathbf{W}_q) - 1 = \frac{1}{d^q}\sum_{j=1}^{d^q}\big(Z_j^2 - 1\big),$$

where each $(Z_j^2 - 1)$ is centered sub-Weibull with parameter $1/q$, we get

$$\mathbb{P}\Big(\Big|\frac{1}{d^q}s_1^2(\mathbf{u}\mathbf{u}^\top\mathbf{W}_1\dots\mathbf{W}_q) - 1\Big| \geq u\Big) = \mathbb{P}\Big(\Big|\frac{1}{d^q}\sum_{j=1}^{d^q}(Z_j^2 - 1)\Big| \geq u\Big)$$

$$\leq 2\exp\big[-C'd^q\min(u^2, u^{1/q})\big],$$

where we have absorbed the dependency on $K = \|(Z_j^2 - 1)\|_{\psi_{1/q}}$ into $C'$. Combining the above with the simple fact that $|z - 1| \geq t$ implies $|z^2 - 1| \geq \max(t, t^2)$, we obtain for any $t \geq 0$ that

$$\mathbb{P}\Big(\Big|\frac{1}{d^{q/2}}s_1(\mathbf{u}\mathbf{u}^\top\mathbf{W}_1\dots\mathbf{W}_q) - 1\Big| \geq t\Big)$$

$$\leq \mathbb{P}\Big(\Big|\frac{1}{d^q}s_1^2(\mathbf{u}\mathbf{u}^\top\mathbf{W}_1\dots\mathbf{W}_q) - 1\Big| \geq \max(t, t^2)\Big)$$

$$\leq 2\exp\big[-C'd^q\min(t^2, t^{2/q})\big],$$

i.e. $s_1(\mathbf{u}\mathbf{u}^\top\mathbf{W}_1\dots\mathbf{W}_q)$ is sub-Weibull with parameter $2/q$ and

$$\Big|\frac{s_1(\mathbf{u}\mathbf{u}^\top\mathbf{W}_1\dots\mathbf{W}_q)}{d^{q/2}} - 1\Big| < t$$

holds with probability at least $1 - 2\exp\big[-C'd^q\min(t^2, t^{2/q})\big]$, i.e. with *overwhelming* probability (Tao, 2012, Definition 1.1.2) □

**Lemma 2.** *Consider $p$ Random Markov matrices $\mathbf{A}_1, \dots, \mathbf{A}_p \in \mathbb{R}^{T\times T}$ as defined in 1.1, and let $\mathbf{1}_{T\times T}$ be the matrix full of ones. Then, almost surely,*

$$s_1\Big(\mathbf{A}_p\dots\mathbf{A}_1 - \frac{1}{T}\mathbf{1}_{T\times T}\Big) = O(T^{-p/2}) \tag{23}$$

*Proof.* Let us first show $s_1(\mathbf{A}_p\dots\mathbf{A}_1) \xrightarrow{a.s.} 1$, as $T$ grows. Each matrix $\mathbf{A}_i$ can be written as the row-normalisation of a table $\mathbf{M}_i$ of i.i.d. random variables, i.e. $\mathbf{A}_i := \mathbf{D}_i\mathbf{M}_i$, where $\mathbf{D}_i$ is a $T \times T$ diagonal matrix containing the inverse row sums of $\mathbf{M}_i$. The entries in $\mathbf{M}_i$ have a finite fourth moment, and, without loss of generality, mean 1 and variance $\sigma^2$. Thus,

$$s_1(T^{p/2}\mathbf{A}_p\dots\mathbf{A}_1) = s_1(T^{p/2}\mathbf{D}_p\mathbf{M}_p\dots\mathbf{D}_1\mathbf{M}_1)$$

$$\leq s_1(T\mathbf{D}_p)s_1(T^{-1/2}\mathbf{M}_p)\dots s_1(T\mathbf{D}_1)s_1(T^{-1/2}\mathbf{M}_1).$$

Following the argument given in Bordenave et al. (2011), $s_1(T\mathbf{D}_i) = 1 + o(1)$ and $s_1(T^{-1/2}\mathbf{X}_i) \leq \sqrt{T} + O(1)$, for all $1 \leq i \leq p$. Therefore,

$$s_1(T^{p/2}\mathbf{A}_p\dots\mathbf{A}_1) \leq \big(\sqrt{T} + O(1)\big)^p(1 + o(1))$$

$$\leq T^{p/2}(1 + o(1)),$$

which yields, almost surely, $\lim s_1(\mathbf{A}_p\dots\mathbf{A}_1) \leq 1$. The converse inequality is an immediate consequence of the closure of the set of Random Markov matrices under matrix multiplication, which gives $\lambda_1(\mathbf{A}_p\dots\mathbf{A}_1) = 1$, combined with $s_1(\mathbf{A}_p\dots\mathbf{A}_1) \geq |\lambda_1(\mathbf{A}_p\dots\mathbf{A}_1)|$. Hence, almost surely, $\lim s_1(\mathbf{A}_p\dots\mathbf{A}_1) = 1$.

Let $\varphi \in \mathbb{R}^T$ be the unit vector such that $\frac{1}{T}\mathbf{1}_{T\times T} = \varphi\varphi^\top$, i.e. $\varphi = T^{-1/2}(1, \dots, 1)^\top$. Also, let $\mathbf{A} := \mathbf{A}_p\dots\mathbf{A}_1$ and define $\mathbf{A}^\perp := \mathbf{A} - \varphi\varphi^\top$. Since the rows of $\mathbf{A}$ sum to 1, our construction ensures that those of $\mathbf{A}^\perp$ sum to zero. We want to show that $s_1(\mathbf{A}^\perp) = s_2(\mathbf{A})\big(1 + o(1)\big)$. To this end, consider the SVD of the matrix $\mathbf{A}^\perp$. There exist orthogonal matrices $\mathbf{U}, \mathbf{V}$ and a diagonal matrix $\mathbf{\Sigma} := \text{diag}\big(s_1(\mathbf{A}^\perp), \dots, s_n(\mathbf{A}^\perp)\big)$ such that

$$\mathbf{A}^\perp = \mathbf{U}\mathbf{\Sigma}\mathbf{V}^\top.$$

Note that since $\mathbf{A}^\perp\varphi = \mathbf{0}$, the matrix has rank at most $T - 1$ and thus $s_n(\mathbf{A}^\perp) = 0$. We will now try to relate the singular values of $\mathbf{A}^\perp$ to those of $\mathbf{A}$, observing that $\mathbf{A}$ is a rank-one perturbation of $\mathbf{A}^\perp$, i.e.

$$\mathbf{A} = \varphi\varphi^\top + \mathbf{A}^\perp$$

$$= \varphi\varphi^\top + \mathbf{U}\mathbf{\Sigma}\mathbf{V}^\top.$$

The squared singular values of $\mathbf{A}$ are exactly the eigenvalues of

$$\mathbf{A}\mathbf{A}^\top = \varphi\varphi^\top + \mathbf{U}\mathbf{\Sigma}^2\mathbf{U}^\top. \tag{24}$$

Since eigenvalues are invariant under orthogonal operators, we can multiply on the left and right by, respectively, $\mathbf{U}^\top$ and $\mathbf{U}$ to get a diagonal matrix perturbed by a rank-one matrix:

$$\mathbf{U}^\top\mathbf{A}\mathbf{A}^\top\mathbf{U} = \mathbf{U}^\top\varphi\varphi^\top\mathbf{U} + \mathbf{\Sigma}^2. \tag{25}$$

Taking the trace, we have

$$s_1^2(\mathbf{A}) + \cdots + s_n^2(\mathbf{A}) = 1 + s_1^2(\mathbf{A}^\perp) + \cdots + s_{n-1}^2(\mathbf{A}^\perp). \tag{26}$$

On the other hand, we can apply Thompson-Lidskii's interlacing inequalities Thompson (1976) on Equation equation 25 to get

$$s_1^2(\mathbf{A}) \geq s_1^2(\mathbf{A}^\perp) \geq s_2^2(\mathbf{A}) \geq s_2^2(\mathbf{A}^\perp) \geq \cdots \geq s_{n-1}^2(\mathbf{A}^\perp) \geq s_n^2(\mathbf{A}) \geq 0. \tag{27}$$

Combining Equations equation 26 and equation 27, one obtains

$$s_1^2(\mathbf{A}) + s_2^2(\mathbf{A}) \geq 1 + s_1^2(\mathbf{A}^\perp).$$

As established earlier, almost surely, $\lim s_1(\mathbf{A}) = 1$. So we conclude that in the limit, almost surely, $s_2(\mathbf{A}) \geq s_1(\mathbf{A}^\perp)$. The converse is already given by equation 27. Therefore we have

$$s_1(\mathbf{A}^\perp) = s_2(\mathbf{A})\big(1 + o(1)\big),$$

almost surely. Note that the same reasoning is valid for the case $p = 1$, and results in $s_1(\mathbf{A}_i^\perp) \xrightarrow{a.s.} s_2(\mathbf{A}_i)$ for any $i$.

Having shown the convergence of the largest singular value of $\mathbf{A}^\perp$ to the second largest singular value of $\mathbf{A}$, we now show that $s_2(\mathbf{A})$ is of order $T^{-p/2}$. To this end, note that the matrix can be written as a rank-one perturbation of the product of $\mathbf{A}_i^\perp$'s, i.e.

$$\begin{aligned} \mathbf{A} &= \mathbf{A}_p \ldots \mathbf{A}_1 \\ &= (T^{-1}\mathbf{1}_{T \times T} + \mathbf{A}_p^\perp) \ldots (T^{-1}\mathbf{1}_{T \times T} + \mathbf{A}_1^\perp) \\ &= T^{-1}\mathbf{1}_{T \times T}\big(\mathbf{I} + \mathbf{A}_1^\perp + \cdots + \mathbf{A}_{p-1}^\perp \ldots \mathbf{A}_1^\perp\big) + \mathbf{A}_p^\perp \ldots \mathbf{A}_1^\perp, \end{aligned}$$

where some of the terms vanish since $\mathbf{A}_i^\perp\varphi = \mathbf{0}$. Given that $\mathrm{rank}(\mathbf{A} - \mathbf{A}_p^\perp \ldots \mathbf{A}_1^\perp) = 1$, we can apply Thompson-Lidskii's inequality to get

$$s_1(\mathbf{A}_p^\perp \ldots \mathbf{A}_1^\perp) \geq s_2(\mathbf{A}).$$

By submutiplicativity of the operator norm, this implies $s_1(\mathbf{A}_p^\perp) \ldots s_1(\mathbf{A}_1^\perp) \geq s_2(\mathbf{A})$. Moreover, we previously established that for each individual matrix $\mathbf{A}_i$, $s_1(\mathbf{A}_i^\perp) \xrightarrow{a.s.} s_2(\mathbf{A}_i)$, and it is shown in Bordenave et al. (2011) that $s_2(\mathbf{A}_i) \xrightarrow{a.s.} 2\sigma T^{-1/2}$. Therefore, we conclude that

$$s_2(\mathbf{A}) \leq \big(2\sigma T^{-1/2}\big)^p = O(T^{-p/2}).$$

Combined with Equation equation A.2, we have

$$s_1(\mathbf{A} - \frac{1}{T}\mathbf{1}_{T \times T}) = s_1(\mathbf{A}^\perp) = O(T^{-p/2}),$$

almost surely. □

**Theorem 9.** *Let $\mathbf{A}_1, \ldots, \mathbf{A}_p \in \mathbb{R}^{T \times T}$ be independent Random Markov matrices as defined in 1.1 and $\mathbf{W}_1 \in \mathbb{R}^{T \times d}$, $\mathbf{W}_2, \ldots, \mathbf{W}_q \in \mathbb{R}^{d \times d}$ be independent Gaussian matrices with i.i.d. $\mathcal{N}(0, 1)$ entries. Then, for large enough $T$ and $d$ with fixed $\gamma = T/d \in (0, 1]$, the event*

$$\left|\frac{s_1(\mathbf{A}_p \ldots \mathbf{A}_1\mathbf{W}_1 \ldots \mathbf{W}_q)}{d^{q/2}} - 1\right| < t,$$

*holds with overwhelming probability.*

*Proof.* We write $\mathbf{A} := \mathbf{A}_p \ldots \mathbf{A}_1 = \varphi\varphi^\top + \mathbf{A}^\perp$ and $\mathbf{W} := \mathbf{W}_1 \ldots \mathbf{W}_q$. Then, using the triangle inequality $|s_1(A) - s_1(B)| \leq s_1(A + B) \leq s_1(A) + s_1(B)$, we have

$$|s_1(\varphi\varphi^\top \mathbf{W}) - s_1(\mathbf{A}^\perp \mathbf{W})| \leq s_1(\mathbf{A}\mathbf{W}) = s_1(\varphi\varphi^\top \mathbf{W} + \mathbf{A}^\perp \mathbf{W})$$
$$\leq s_1(\varphi\varphi^\top \mathbf{W}) + s_1(\mathbf{A}^\perp \mathbf{W}).$$

On the other hand, it is well known that the largest singular value of a Gaussian matrix converges almost surely to the soft edge of the bulk of the limiting density Geman (1980), i.e.

$$s_1\left(\frac{1}{\sqrt{d}}\mathbf{W}_i\right) \xrightarrow{a.s.} \begin{cases} 1 + \sqrt{\gamma}, & i = 1, \\ 2, & i \geq 2. \end{cases}$$

Therefore, by submultiplicativity of $s_1$, we have

$$s_1(\mathbf{W}) \leq s_1(\mathbf{W}_1) \ldots s_1(\mathbf{W}_q) \leq \left(2\sqrt{d} + o(\sqrt{d})\right)^q = 2^q d^{q/2} + o(d^{q/2}). \tag{28}$$

Combining equation 28 with Lemma 2, we get

$$s_1(\mathbf{A}^\perp \mathbf{W}) \leq s_1(\mathbf{A}^\perp)s_1(\mathbf{W}) = O(d^{\frac{q-p}{2}}), \tag{29}$$

and thus, almost surely,

$$\left|s_1(\varphi\varphi^\top \mathbf{W}) - O(d^{\frac{q-p}{2}})\right| \leq s_1(\mathbf{A}\mathbf{W}) \leq s_1(\varphi\varphi^\top \mathbf{W}) + O(d^{\frac{q-p}{2}}).$$

Now, using Lemma 1, we can assert that $s_1(\varphi\varphi^\top \mathbf{W})$ is close to $d^{q/2}$ with overwhelming probability, i.e.

$$\frac{s_1(\varphi\varphi^\top \mathbf{W})}{d^{q/2}} \in (1 - t, 1 + t),$$

with a probability greater than $P_{t,d} := 1 - 2\exp\left[-C'd^q \min(t^2, t^{2/q})\right]$. Moreover, by equation 29,

$$\frac{s_1(\mathbf{A}^\perp \mathbf{W})}{d^{q/2}} \to 0,$$

as $d$ grows. Thus, we can make the above quantity smaller than any given $\varepsilon$. Altogether, for large enough $T$ and $d$, the probability that

$$\left|\frac{s_1(\mathbf{A}\mathbf{W})}{d^{q/2}} - 1\right| < t + \varepsilon$$

is at least $P_{t,d}$. Since $\varepsilon$ is arbitrary the proof is complete.

$\square$

**Theorem 10.** *Let $\mathbf{A}_1, \ldots, \mathbf{A}_p \in \mathbb{R}^{T \times T}$ be Random Markov matrices as defined in 1.1 and $\mathbf{W}_1 \in \mathbb{R}^{T \times d}$, $\mathbf{W}_2 \ldots, \mathbf{W}_q \in \mathbb{R}^{d \times d}$ be independent Gaussian matrices with i.i.d. $\mathcal{N}(0,1)$ entries. Then, for $T$ and $d$ large enough,*

$$s_2(\mathbf{A}_p \ldots \mathbf{A}_1 \mathbf{W}_1 \ldots \mathbf{W}_q) = O(d^{\frac{q-p}{2}}). \tag{30}$$

*Proof.* To exhibit a spectral gap in $\mathbf{A}\mathbf{W}$, it suffices to bound its second largest singular value by a quantity significantly lower than where the largest singular value is concentrated. To this end, observe that $\mathbf{A}\mathbf{W}$ is a rank-one perturbation of $\mathbf{A}^\perp \mathbf{W}$:

$$\mathbf{A}\mathbf{W} = (\mathbf{A}^\perp + \varphi\varphi^\top)\mathbf{W} = \mathbf{A}^\perp \mathbf{W} + \varphi\varphi^\top \mathbf{W}.$$

Thus, using Weyl's inequality, we can write

$$s_2(\mathbf{A}\mathbf{W}) \leq s_1(\mathbf{A}^\perp \mathbf{W}) + s_2(\varphi\varphi^\top \mathbf{W}) = s_1(\mathbf{A}^\perp \mathbf{W}).$$

Next, by submultiplicativity of the operator norm combined with upper bounds in Lemma 2 and equation 28,

$$s_1(\mathbf{A}^\perp \mathbf{W}) \leq s_1(\mathbf{A}^\perp)s_1(\mathbf{W}) = O(T^{-p/2})O(d^{q/2}).$$

Therefore,

$$s_2(\mathbf{A}\mathbf{W}) = O(d^{\frac{q-p}{2}}).$$

$\square$

**Lemma 3** (Bulk distribution of $\mathbf{A}^\perp$). *Let $\mathbf{A} \in \mathbb{R}^{T \times T}$ be a Random Markov matrix, and let $\mathbf{A}^\perp :=$ $\mathbf{A} - T^{-1}\mathbf{1}_{T \times T}$. Then, almost surely, the empirical singular value distribution of $T^{1/2}\mathbf{A}^\perp$ weakly converges to the quartercircular law as $T \to \infty$, i.e.*

$$\nu_{\sqrt{T}\mathbf{A}^\perp} := \frac{1}{T} \sum_{i=1}^{T} \delta_{s_i(\sqrt{T}\mathbf{A}^\perp)} \xrightarrow{\mathcal{C}_b} \mathcal{Q}_\sigma, \tag{31}$$

*where $\mathcal{Q}_\sigma$ is the quartercircular law on the real interval $[0, 2\sigma]$ with Lebesgue density*

$$x \mapsto \frac{1}{\pi\sigma^2}\sqrt{4\sigma^2 - x^2}\mathbb{1}_{[0,2\sigma]}.$$

*Moreover, almost surely, $\mathbf{A}^\perp$ does not exhibit any spectral gap.*

*Proof.* Thompson-Lidskii's interlacing result for finite rank perturbation Thompson (1976) states that for any $n \times n$ matrices $\mathbf{M}$ and $\mathbf{M}'$ with $\mathrm{rank}(\mathbf{M} - \mathbf{M}') \le k$, we have

$$s_{i-k}(\mathbf{M}) \le s_i(\mathbf{M}') \le s_{i+k}(\mathbf{M}).$$

This in turn yields the following bulk inequality,

$$\|F_\mathbf{M} - F_{\mathbf{M}'}\|_\infty \le \frac{\mathrm{rank}(\mathbf{M} - \mathbf{M}')}{n},$$

where $F_\mathbf{M}$ and $F_{\mathbf{M}'}$ denote the cumulative distribution functions of $\nu_\mathbf{M}$ and $\nu_{\mathbf{M}'}$, respectively. Since $\mathrm{rank}(\mathbf{A} - \mathbf{A}^\perp) = 1$, then

$$\|F_{\sqrt{T}\mathbf{A}} - F_{\sqrt{T}\mathbf{A}^\perp}\|_\infty \le \frac{1}{T} \xrightarrow[T \to \infty]{} 0.$$

Combining the above limit with the fact that $\nu_{\sqrt{T}\mathbf{A}} \xrightarrow{\mathcal{C}_b} \mathcal{Q}_\sigma$ almost surely (see Bordenave et al. (2011)), we deduce that

$$\nu_{\sqrt{T}\mathbf{A}^\perp} \xrightarrow{\mathcal{C}_b} \mathcal{Q}_\sigma$$

almost surely. The almost sure absence of outliers in the singular value distribution of $\mathbf{A}^\perp$ can be immediately inferred from Lemma 2 when $p = 1$. $\qquad\square$

**Lemma 4.** *Let $0 < \sigma_i < \infty$ and $0 < \gamma_i \le 1$ for $1 \le i \le n$. Let $\mathcal{M}$ be the free multiplicative convolution of $\mathcal{MP}(\gamma_i, \sigma_i)$ distributions, i.e.*

$$\mathcal{M} := \mathcal{MP}(\gamma_1, \sigma_1) \boxtimes \mathcal{MP}(\gamma_2, \sigma_2) \boxtimes \cdots \boxtimes \mathcal{MP}(\gamma_n, \sigma_n).$$

*Then the mean and variance of $Z \sim \mathcal{M}$ are given by*

$$\mathbb{E}(Z) = \prod_{i=1}^{n} \sigma_i^2, \tag{32}$$

$$\mathrm{Var}(Z) = \Big(\prod_{i=1}^{n} \sigma_i^2\Big)^2 \big(\gamma_1 + \gamma_1\gamma_2 + \cdots + \gamma_1\gamma_2 \cdots \gamma_n\big). \tag{33}$$

*Proof.* The distribution in question $\mathcal{M}$ is the limiting squared singular value distribution of a product of rectangular independent Gaussian matrices, whose general moments are worked out in (Akemann et al., 2013, equation 58). Simple algebraic manipulations lead to our result. $\qquad\square$

## A.3 SUPPLEMENTARY DATA

### A.3.1 IMPLEMENTATION DETAILS

**Architecture.** The default model consists of a stack of single-head attention layers, with an optional LayerNorm inserted between them (denoted by "+ LN" in the legend) after receiving an optional skip connection from the previous layer (denoted by "+ skip" in the legend). When both options are enabled simultaneously, the configuration is referred to as "+ skip + LN". By single-head, we mean that only one attention mechanism is computed, applied to the values and then multiplied by a matrix $\mathbf{W}_h$ which is initialised as the identity matrix and optimised during training.

**Attention design.** At initialisation, when the attention is labelled as "$\mathbf{A}$", the matrix is sampled from the set of Random Markov matrices, as defined in Definition 1.1, with a variance of $\sigma_A = 1$. To achieve this, we sample a random matrix $\mathbf{B}$ with i.i.d. lognormal entries and apply softmax row-wise such that $\mathbf{A} := \operatorname{softmax}(\mathbf{B})$. The moments of $\mathbf{B}$ are adjusted precisely so that $\sigma_A = 1$. During training, the entries of $\mathbf{B}$ are optimised. If "Identity $\mathbf{A}$" is chosen, the attention matrix is a constant equal to the identity only at initialisation and then optimised at training time. When the attention is labelled as "$\mathbf{A}(\mathbf{X})$", the key/query matrices are sampled from i.i.d. Gaussian matrices $\mathcal{N}(0,1)$ and the standard key-query attention matrix is formed. If a mention to "Xavier" appears in the legend, it means the key-query matrices are sampled from a rescaled Gaussian $\mathcal{N}(0, d_{qk}^{-1})$. Updates are performed on $\mathbf{W}^Q$ and $\mathbf{W}^K$. If the label indicates a "$\perp$", the forward pass of the attention mechanism is systematically (at initialisation and for all following training steps) adjusted so that the spectral gap is removed, as in our modified model (9).

**Training.** Given some isometric $\mathbf{X}_0$ input data, the goal is to learn the entrywise Heaviside function, a non-trivial task due to the function's nonlinearity. To achieve this, we train a series of attention-only transformer encoders on a mean squared error (MSE) loss, optimised with Adam. We conduct an extensive grid search over the learning rate $\lambda \in \{1,3,5\} \times 10^{-\{1,2,3,4,5\}}$. Each experiment is run 5 times, and the learning rate that results in the best average training performance for each configuration, as shown in the plots, is selected. The figures display the training loss with respect to training steps, i.e. the number of gradient descent updates. A "no training" label is shown when no training progress is made after 50 training epochs, despite tuning the learning rate. We train on a set of 50 data points, each of size $T \times d$, with $T = d = 500$ to ensure we are in the large width regime that our theoretical framework presupposes.

### A.3.2 Additional experiments

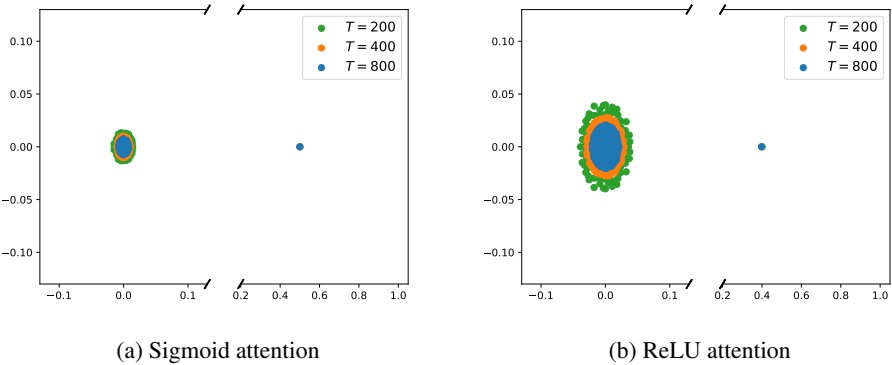

(a) Sigmoid attention        (b) ReLU attention

Figure 9: Some variants of the attention mechanism proposed in the literature apply different activation functions such as (a) sigmoid or (b) ReLU on the key-query dot products; see Wortsman et al. (2023). Similar to the softmax-based attention (Figure 1), the spectra of these alternative attention matrices also display an outlier. Interestingly, the technical framework developed in our paper can be applied to analyse the signal propagation on sigmoid- or ReLU-based transformers.

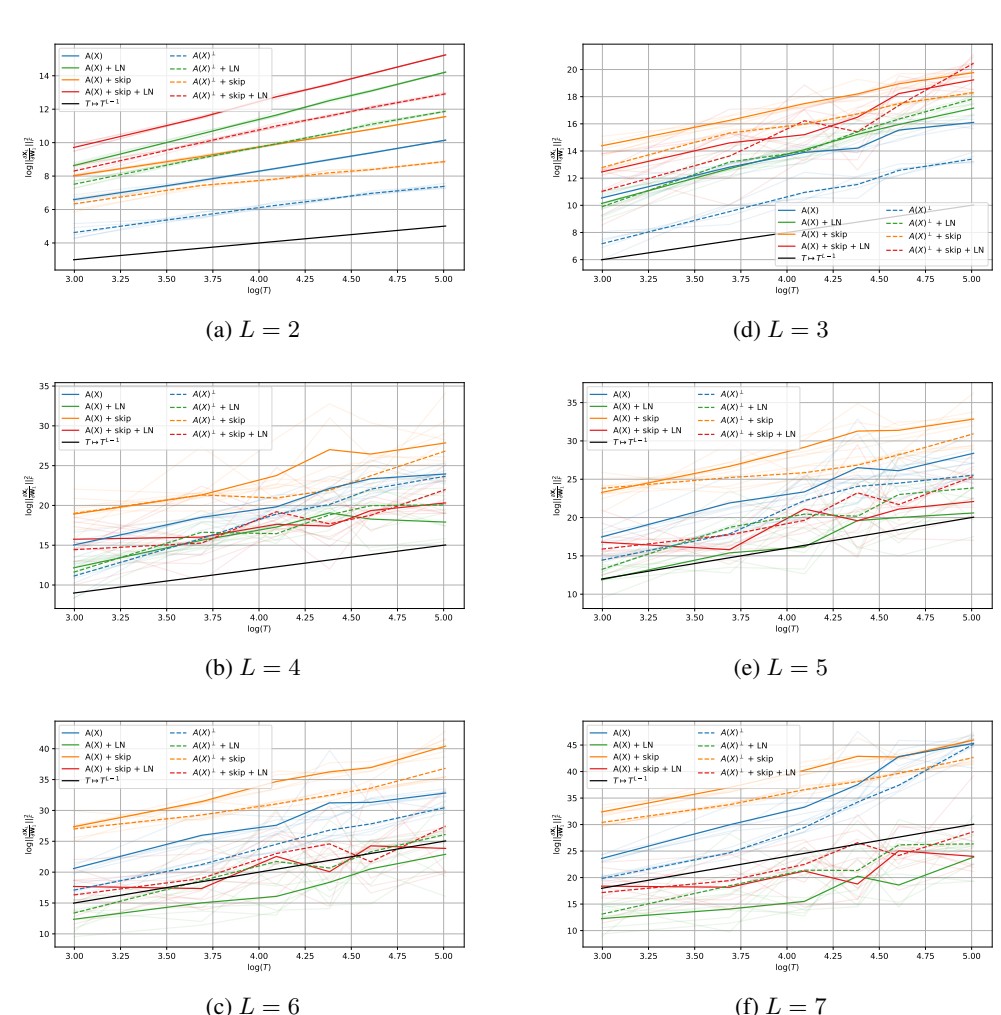

(a) $L = 2$         (d) $L = 3$

(b) $L = 4$         (e) $L = 5$

(c) $L = 6$         (f) $L = 7$

Figure 10: In a transformer with key-query attention, the gradient norm explodes in width at a rate that worsens with increasing depth $L$, exceeding the growth of $T^{L-1}$ predicted by our analysis.

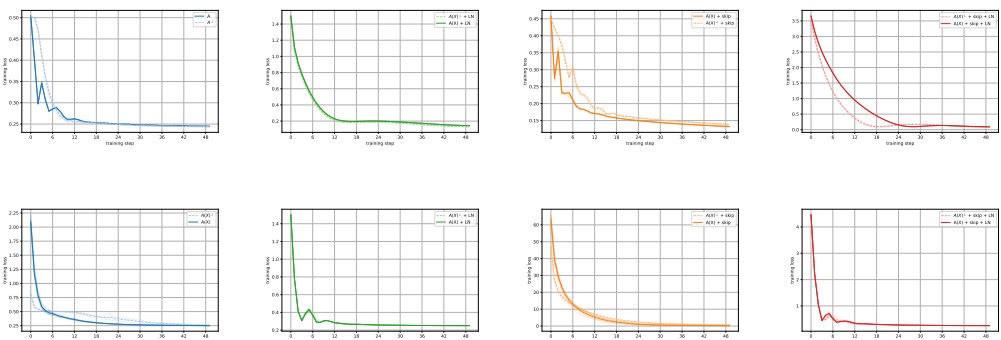

Figure 11: Examples of training loss curves with and without removing the gap for single-layer (top row) and two-layer (bottom row) transformers.

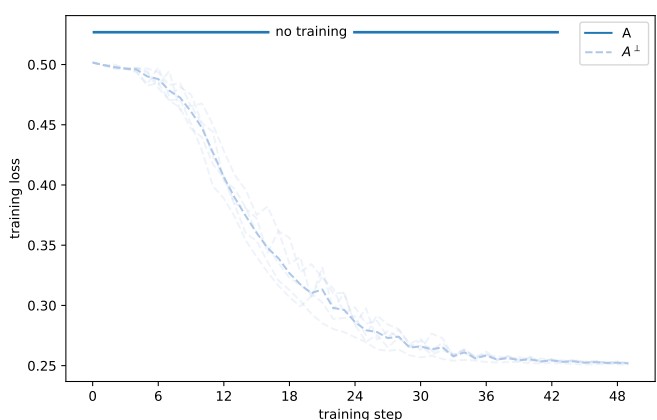

Figure 12: An example of how rank collapse can inhibit the training of a 5-layer transformer. Our fix is proposed to eliminate the main cause of rank collapse, effectively reducing the possibility of a "no training" scenario in this situation; see section A.3 for implementation details.

