# OpenReview forum: "Mind the Gap: a Spectral Analysis of Rank Collapse and Signal Propagation in Transformers"
_ICLR.cc/2025/Conference — Submitted to ICLR 2025_

### Official Review · Reviewer_smfh · 2024-11-01

**Soundness:** 3
**Presentation:** 2
**Contribution:** 2
**Rating:** 5
**Confidence:** 3

**Summary:**

The manuscript studies the self-attention mechanism at initialization through free probability and random matrix theory. Free probability deals with non-commuting random variables such as random matrices, making it a good fit for the analysis of attention matrices in transformers. The attention matrices are usually large due to numerous tokens, thus the corresponding spectral properties become predictable. In particular, the eigenvalues of softmax-attention matrix at the first layer are similar to the ones of the random Markov matrix - there is an outlier eigenvalue that equals one while the remaining eigenvalues form a bulk centered at zero with radius $T^{-0.5}$. The manuscript then links this discrepancy in eigenvalues, referred to as spectral gap, with rank collapse and exploding gradients. Fixing these issues could improve the optimization. Thus, the manuscript proposes a solution that adapts the softmax-attention matrix in order to remove the spectral gap. Namely, 1/T is subtracted from each element of the square attention matrix, where T is the matrix dimension. This solution is validated in toy experiments with multi-layer transformers. The experiments indicate that the proposed solution is effective in slowing rank collapse when the spectrum contains a single outlier.

**Strengths:**

S1. Viewing softmax-attention matrices as random Markov matrices and analysing them via free probability seems like a sound approach.

S2. Explaining rank collapse and exploding gradient with the spectral gap (i.e. discrepancy in eigenvalues) is an interesting takeaway.

S3. The proposed solution for mitigating the spectral gap is convenient and easy to implement.

S4. Experiments indicate that the proposed solution slows the rank collapse.

**Weaknesses:**

W1. The proposed analysis holds only for the first layer attention matrix. This limits the usefulness of the proposed analysis. Could the proposed analysis incorporate the attention matrices in deeper layers? What are the key issues?

W2. Removing the spectral gap does not benefit optimization in transformers with two layers, as indicated by the bottom row of Figure 6. What happens in the case of even deeper transformers? Does the removal of the spectral gap have any practical relevance?

W3. Figure 8 and the corresponding text raise questions about the validity of Xavier's initialization. However, this analysis is conducted only for a single-layer transformer (line 488). Can we observe the same issues with initialization in deeper transformers?

**Questions:**

See weaknesses.

---

> ### Author Response · Authors · 2024-11-19
>
> We thank the reviewer for their careful comments and appreciate their recognition of our work's "sound'' and "interesting'' approach. We want to add a few points regarding the listed weaknesses:
> 1. Our Markov matrix model is used because of its known limiting spectral distribution; see Theorem 2. Although it captures the attention matrix in the first layer only, the idea of studying the spectrum of attention matrices at initialisation can certainly be extended to deeper layers as well. In fact, we are currently working on generalising our analysis by allowing more complicated Markov matrix models to model attention in deeper layers. The main issue would be to relax the assumption of independence of pre-normalisation attention entries and still obtain an analogue of Theorem 2. Such an analysis is complicated by the need to develop new results in random matrix theory about this broader class of random Markov matrices.
> 2. Thanks to your comment, we have now relegated the previous Figure 6 to the appendix as we are convinced it was misleading. To clarify, we do not make any claim about the possible benefits of our fix to the "training accuracy'' once training does take place. It is the very "trainability'', i.e. the success rate of the training process in converging to any reasonable local min, which is at risk in the presence of rank collapse and exploding gradients at initialisation. We have now added Figure 12 which shows how our proposed fix enables training in a simple case (whilst without it no training occurs), leaving a large-scale verification of its effectiveness in real-world transformers for future investigations. We, however, included in our related work section pointers to empirical works exploring this idea, such as [1], [2], and [3].
> 3. We appreciate your meticulous reading and raising of this interesting point. First, we wish to emphasise that the soundness (or lack thereof) of a certain scaling depends on the scaling of the input signal. It is only under the assumption of $\mathbf{X}_0 \mathbf{X}_0^\top = \mathbf{I}$ that we show that Xavier's initialisation of key and query matrices leads to degenerate attention. Therefore, wherever Xavier is used in practice, the input covariance matrix is scaled differently, e.g. $\mathbf{X}_0 \mathbf{X}_0^\top \approx T\mathbf{I}$. Now, back to your question, if we stick to our assumption of isometric input, the same degenerate behaviour is also observed in deeper layers. That is because Xavier initialisation generates uniform attention at the first layer $\mathbf{A}_1 = \frac{1}{T} \mathbf{1}_T$ which forces $\mathbf{X}_1 = \mathbf{A}_1 \mathbf{X}_0 \mathbf{W}^{V}_1  = \frac{1}{T}\mathbf{1}_T \mathbf{X}_0 \mathbf{W}^{V}_1$ to be rank-one. Considering the key-query products at the second layer $$\frac{\mathbf{X}_1 \mathbf{W}^{Q}_2 {\mathbf{W}^{K}_2}^\top \mathbf{X}_1^\top}{\sqrt{d_q}},$$ one observes that the problem is all the more exacerbated. Not only $\mathbf{X}_1$ is not scaled up compared to $\mathbf{X}_0$, but also it is projected down to one dimension.
>
> -----
>
> [1] Ali, A. et al. (2023). Centered self-attention layers.
>
> [2] Noci, L. et al. (2024). The shaped transformer: Attention models in the infinite depth-and-width limit.
>
> [3] Ye, T. et al. (2024). Differential transformer.

---

> > ### Comment · Reviewer_smfh · 2024-11-26
> > **Comments acknowledged**
> >
> > I have reviewed the authors' comments and appreciate their responses. I encourage the authors to continue exploring the applications of the Markov matrix model to deeper attention layers as well as large-scale validation of the proposed theory. Such advancements would significantly enhance the manuscript's contributions. However, based on the current state of the manuscript, I have decided to maintain my original score.

---

> > > ### Author Response · Authors · 2024-11-26
> > >
> > > We appreciate the reviewer for engaging with our work and providing their perspective. However, there appears to be a misunderstanding. We completely agree with them that adding large scale experiments and rigorously analysing deeper layers would significantly improve not only this paper but have a high immediate impact on the whole field (most likely elevating this submission to a spotlight or oral paper). However, our goal is not to persuade the reviewer to rate this work as a 10 or 8 but rather to recognise that the current submission already provides substantial insights and contributions to the community to warrant acceptance (i.e., a score above 5, 5 being a score that leans towards rejection).
> > >
> > > Whilst the additional dimensions raised by the reviewer are valuable (this is precisely why we openly mentioned those two points as natural follow up works in our paper), they are also highly non-trivial (and we give a sense of why in our work). For instance, large-scale experiments often merit standalone papers (e.g., see [1]), and determining the limiting spectral behavior of attention matrices in deeper layers could easily constitute a whole dedicated PhD thesis in random matrix theory. Let us emphasise once again that our work leverages cutting edge results from random matrix theory—achieved through years of collaboration among multiple researchers—to present a cutting-edge analysis of Transformer's initialisation theory. This led to novel findings, such as rank collapse in width, which were previously unknown. Requiring our submission to include large scale experiments on real world problems, in addition to coming up with new random matrix theory findings, on top of our already existing substantial contributions to warrant acceptance (i.e. score > 5) at this venue is not aligned with ICLR's guidelines (and probably an unrealistic ambition for a single paper).
> > >
> > > Indeed, citing ICLR's reviewer guidelines: *“Does the work contribute new knowledge and sufficient value to the community? Note, this does not necessarily require state-of-the-art results. Submissions bring value to the ICLR community when they convincingly demonstrate new, relevant, impactful knowledge (incl., empirical, theoretical, for practitioners, etc.).”* We firmly believe and reaffirm here that our submission meets these criteria by bringing value to the community (at least the one interested in coming up with a theory of *why* and *how* Transformers work) and provides enough value to warrant acceptance (score > 5) in its current form.
> > >
> > >
> > > Finally, we also note that the review might unfortunately be primarily focusing on aspects absent from the paper rather than engaging with the content and contributions it already offers. To ensure clarity and address any potential oversight, we will briefly summarize our key contributions below:
> > >
> > >
> > > - We introduce a mathematical framework to analyse transformers based on free probability and random matrix theory, that can open avenues for further theoretical developments.
> > > - This framework proved effective in our case as we could (i) recover previously known phenomena like vanishing gradients and rank collapse in depth (Propositions 3 and 4), (ii) unveil for the first time the phenomenon of rank collapse in width (Proposition 3), (iii) quantify exactly how the latter exacerbates the former (Proposition 3), (iv) propose a simple fix that comes across as natural through this new perspective (Section 2.2), and (v) show its positive impacts on the information flow in our transformer model at initialisation (Propositions 5 and 6). Proposition 1 demonstrates how an attention layer fits exactly within our theoretical framework with no further modification, highlighting its practical relevance.
> > >  - All statements are rigorously proven and empirically verified, making a strong case for the set of techniques used in our proofs to be useful and to inspire other researchers in the field.
> > > - Our experiments also explore the potential universality of our findings when more modules like layer normalisation or residual connections are integrated into the transformer block (Figure 4). State-of the-art Transformers like BERT are shown to suffer from both rank collapse in width and depth (Figure 6). At the same time, we transparently address the limitations of our study, offering insights into why generalising our analysis to deeper layers would require non-trivial efforts (Figure 7) that we hope this foundational work will facilitate.
> > >
> > > We hope the reviewer will revise their score in light of this note and thank them once again for their time.
> > >
> > > -----
> > >
> > > [1] Ye, T. et al. (2024). Differential transformer.

---

### Official Review · Reviewer_Htgk · 2024-11-03

**Soundness:** 3
**Presentation:** 3
**Contribution:** 2
**Rating:** 5
**Confidence:** 3

**Summary:**

- The study reveals that softmax-based attention mechanisms during the initialization of Transformers can cause pathological signal and gradient propagation through the network, manifesting as vanishing/exploding gradients and rank collapse—where all tokens converge to a single representation at depth.
- This paper identifies and analyze rank collapse in width, which is caused by the spectral gap between the two largest singular values in the attention matrix.
- A solution is proposed to address the rank collapse and exploding gradients issues.

**Strengths:**

- The paper is clearly written and well organized
- This paper provides a novel concept named rank collapse in width, advancing understanding of initialization issues. I.e, the gap between the largest eigen value and other eigen value bulk will lead to rank collapse, preventing the processing of input tokens.
- Rigorous analysis is provided about the reasons of gap, how gap leads to randk collapse and exploding gradients.
- A reasonable solution is proposed to deal with the rank collapse, in which, rank-one perturbation is removed from every attention matrix to close the gap. The effectiveness of this solution is analyzed in Proposition 5, 6.
- extensive experiments are provided to support the analysis.

**Weaknesses:**

- The analysis is based on simple model structure with attention-only single-head transformer. Analysis on more complex models with multiple layers, skip connections and layernorm is preferred.
- The paper assumes Gaussian-distributed initialization for parameters, which is not always optimal in practice. Transformers often use Xavier or He initialization, will the results still hold?
- lack empirical validation. Most experiments are conducted to evaluate the rank collapse, gradient exploding and loss curve, there are limited experiments on real datasets and across different tasks.
- The paper analyzes the initialization stage of transformers without considering parameter updating during training. Spectral properties change over training, and over multi-layer transformers, which may significantly affect rank collapse and gradient stability analysis in this paper.

**Questions:**

Does the rank collapse and gradient exploding also exist in the existing pre-trained large language models?

---

> ### Author Response · Authors · 2024-11-19
>
> We thank the reviewer for their helpful comments and support of our rigorous analysis. Regarding the listed weaknesses:
> - We admit that our model does not capture the full complexity of state-of-the-art transformer architectures used in practice. We would like, however, to highlight that analysing even this simplified model posed significant technical challenges. Additionally, the model is sufficiently meaningful to provide relevant insights about the real case. Finally, this work is by no means a conclusive analysis of transformers at initialisation and further extensions are currently being developed, based on the framework set by this manuscript. That said, the harmful consequences of the spectral gap are empirically shown to persist even in the presence of other mitigating architecture designs (see Figure 4); removing this spectral gap has a clear promising benefit.
> - Thank you for raising this point. As we discussed in a remark at the beginning of Section 2, we have chosen (non-scaled) Gaussian initialisation for value matrices based on the fact that softmax attention is essentially a downscaling. In fact, if one initialises the value matrices using Xavier or He scheme, one would get an even faster collapse of the stable rank. In practice, this is mitigated by the use of other modules such as LayerNorm and skip connections as we have discussed at the end of Section 3; see Figure 8b.
> - Our paper has a clear focus on the theory. While we acknowledge the need for further large-scale implementations, we believe such a task falls out of the scope of the current manuscript. Instead, we point the reader to recent results by [1], [2], and [3], where similar adaptations to the attention mechanism are explored with a more experimental focus. [1], in particular, consider the same removal of the leading spectral component and give a substantial experimental section exploring its efficacy. Our theoretical contribution complements their more experimental results. We believe that both the theory and large-scale simulations are substantial enough to warrant separate investigations and acceptance at this venue.
> - As pointed out in our general response, our paper contributes to a body of work concerned with neural networks at initialisation and does not address the training dynamics altogether. Our study, however, has consequences for the success of the training process, i.e. trainability, and can be used to decrease the risk of failed training. This is highlighted in the initial excessive stability of the network as a consequence of rank collapse which implies the network is approximately projecting all inputs onto a one-dimensional subspace. Moreover, the linear growth of the gradient norm with token length $T$ in even a single layer suggests that exploding or vanishing gradients can be an expected consequence of the spectral gap. Both of these unfortunate consequences of the softmax-based attention can be resolved by removing the spectral outliers.  A more comprehensive study of the spectral properties during or after training can be further explored in follow-up works.
>
> Regarding the question:
> - We have added a plot to the main body (Figure 3) to show that famous (untrained) transformers suffer from rank collapse in both width and depth. We believe it to be a great addition to our manuscript and thank the reviewer for raising this point. However, as we discussed in our general response, (pre-)trained networks are not the subject of our study.
>
> ------
> [1] Ali, A. et al. (2023). Centered self-attention layers.
>
> [2] Noci, L. et al. (2024). The shaped transformer: Attention models in the infinite depth-and-width limit.
>
> [3] Ye, T. et al. (2024). Differential transformer.

---

### Official Review · Reviewer_Ua2p · 2024-11-03

**Soundness:** 3
**Presentation:** 3
**Contribution:** 3
**Rating:** 6
**Confidence:** 2

**Summary:**

In this paper, by analyzing the spectral gap in the eigenvalues of the attention matrix, we reveal the rank collapse in width that exists in the classical transformer structure, and point out that this phenomenon leads to the disappearance of the training gradient as well as the rank collapse, which in turn causes the training instability of the transformer, and subsequently The corresponding solutions are given in this paper.

**Strengths:**

1. The paper is logically organized, with a clear theoretical narrative and a high degree of readability
2. This paper approaches the issue of Transformer performance from a relatively rare angle, pointing out that rank collpase occurs not only as the number of layers (“depth”) of the transformer increases, but also as the “width” of the It is pointed out that rank collpase occurs not only with the increase of the number of layers (“depth”) of the transformer, but also with the increase of the “width”, and also leads to the emergence of training instability, and further pointed out the causes and solutions, which is of high theoretical significance.

**Weaknesses:**

1. What is discussed in this paper has strong theoretical implications, but is studied in the context of an extremely simplified Transformer model, and it is not possible to determine whether the phenomenon also exists for various types of Transformer model variants, such as linear transformer. It is also difficult to determine whether the problem also occurs in transformer models that add many tricks to mitigate rank collpase in depth.
2. Taking up the weakness 1, although this paper is of high theoretical significance, it is hoped that the authors' experiments will reflect whether the various Transformers in practical use have similar problems, or, under what circumstances similar problems may affect the results.

**Questions:**

see weakness

---

> ### Author Response · Authors · 2024-11-19
>
> We thank the reviewer for their careful comments and support of both the novelty and clarity of the analysis. The points they raised as weaknesses are addressed at some length in our general response. To summarise:
>
> 1. Thanks to your comment, we have now added a plot (Figure 1c) showing the absence of any spectral gap in linear transformers. Furthermore, although our mathematical statements are proven for a simple model, we have provided empirical evidence that (i) rank collapse (in depth and width) occurs in other variants (Figure 2) and even real-world models (Figure 3), and (ii) the insight drawn from our analysis is indeed relevant for practical purposes. In the related works section, we have included citations to recent empirical studies that suggest a similar "mean subtraction'' method as evidence of our work's relevance and applicability.
>
> 2. We appreciate the reviewer's recognition of the theoretical significance of our work. We have now included plots (Figure 3) demonstrating the occurrence of rank collapse (in depth and width) in real-world architectures. We believe it is a great addition to our manuscript and thank the reviewer for raising this point.
>
> We hope the reviewer is content with our edits and explanations and will consider raising their score.

---

> > ### Comment · Reviewer_Ua2p · 2024-11-27
> >
> > I'm very sorry that my response is so late, I've been made aware of the author's responses to the questions I asked, and I think they appropriately respond to some of the concerns, so I'll keep my rating the same.

---

> > > ### Author Response · Authors · 2024-11-27
> > >
> > > We thank the reviewer for stepping in and are glad they are reaffirming their appreciation for our work and acknowledgement of its "high theoretical significance". We are happy to address any additional concerns to further strengthen the reviewer's confidence in supporting the acceptance of our paper.

---

### Official Review · Reviewer_ALfq · 2024-11-05

**Soundness:** 2
**Presentation:** 2
**Contribution:** 2
**Rating:** 3
**Confidence:** 3

**Summary:**

The paper addresses the issue of rank collapse in softmax attention. While the well-known rank collapse in depth is discussed, the authors also identify a new variant: rank collapse in width. By modeling softmax-based attention at initialization using Random Markov matrices, the paper pinpoints the root cause of the problem as the spectral gap between the two largest singular values of the attention matrices. By eliminating this spectral gap, the issue of rank collapse has been effectively resolved.

**Strengths:**

The finding is interesting and the solution is simple.

**Weaknesses:**

1. I propose that the current introduction section be divided into two distinct parts: an introduction section and a preliminary section. The introduction should encompass the existing situation, motivation, a succinct overview of the proposed solution, and the results obtained. The preliminary section should offer the necessary background knowledge essential for understanding the paper.

2. The phenomenon of width collapse is frequently observed in current transformer training, or it may necessitate specific data conditions to manifest. While transformers have been extensively utilized in domains such as computer vision (CV) and natural language processing (NLP) for several years, reports of width collapse remain absent. Given that the proposed method solely addresses this width collapse, I am interested in its actual efficacy in real-world applications.

3. The proposed solution involves the direct substitution of A with its variant within the attention layer. However, the performance implications of this adjustment have not been comprehensively validated. The author has provided only a comparison of training loss, indicating that this modification may influence the training loss curve. I would appreciate the inclusion of ablation studies conducted on actual transformer models in either language modeling (LM) or CV tasks. For example, in the case of LM, it is essential to supply benchmark results within controlled scenarios. Furthermore, it would be beneficial to understand how this modification may affect both training and inference speed.

4. I understand this paper only investigates the rank collapse in softmax attention transformers. However, I am curious whether this problem exists in other attention variants, such as linear attention.

**Questions:**

As above.

---

> ### Author Response · Authors · 2024-11-19
>
> We appreciate the reviewer’s feedback and would like to highlight the strengths of this paper to ensure they are not overlooked:
> - We introduce a mathematical framework to analyse transformers based on free probability and random matrix theory, that can open avenues for further theoretical developments.
> - This framework proved effective in our case as we could (i) recover previously known phenomena like vanishing gradients and rank collapse in depth (Propositions 3 and 4), (ii) unveil for the first time the phenomenon of rank collapse in width (Proposition 3), (iii) quantify exactly how the latter exacerbates the former (Proposition 3), (iv) propose a simple fix that comes across as natural through this new perspective (Section 2.2), and (v) show its positive impacts on the information flow in our transformer model at initialisation (Propositions 5 and 6). Proposition 1 demonstrates how an attention layer fits exactly within our theoretical framework with no further modification, highlighting its practical relevance.
> - All statements are rigorously proven and empirically verified, making a strong case for the set of techniques used in our proofs to be useful and to inspire other researchers in the field.
> - Our experiments also explore the potential universality of our findings when more modules like layer normalisation or residual connections are integrated into the transformer block (Figure 4). At the same time, we transparently address the limitations of our study, offering insights into why generalising our analysis to deeper layers would require non-trivial efforts that we hope this foundational work will facilitate.
>
> Regarding individual mentioned weaknesses:
> 1. We appreciate your suggestion, however, reviewers Ua2p and Htgk have found the paper logically organised and well-written.
> 2. The phenomenon of rank collapse is indeed observed in real-world transformers at initialisation, as shown in the original paper that identifies rank collapse (in depth); see Figure 2 in [1]. Thanks to your comment, we have added a similar plot (Figure 3) to the paper showing rank collapse in depth and width for a few standard real-world transformers, e.g., BERT and its variants.
> 3. As pointed out in our general response, this paper has a theoretical essence and therefore does not claim nor attempt to yield comprehensive empirical validation across tasks. We believe that the foundational theory developed in this manuscript is sufficient on its own and fits well within the aim and scope of ICLR and would encourage the community to take it further and give it an empirical dimension. Considering all the comments, we decided to move the training accuracy plots to the appendix (Figure 11 in the current version) as they were distracting the reviewers/readers from the primary focus of our work: initialisation. We have also added plots (Figure 3) showing rank collapse, both in width and depth, indeed occurs in real-world architectures and updated the section title to "Experiments and Further Insights". Regarding the inference speed, the reviewer may find it interesting that by controlling the spectra of attention matrices via our fix, the signal entries are also regulated in magnitude. This, in turn, enhances the effectiveness of the quantisation of the signal using fewer bits, as discussed in Section 3.7 of [2].
> 4. Thank you for raising this question. We have added a panel to Figure 1 to showcase the absence of outliers in the spectrum of a linear attention matrix. Figure 9 also exhibits the spectra of other attention variants, e.g. sigmoid- and ReLU-based, described in [3]. Interestingly, these last two exhibit a spectral gap that our theoretical framework could now fully determine, explain and fix.
>
> We hope these clarifications will help the reviewer appreciate our work and revise their score.
>
> --------
>
> [1] Dong, Y. et al. (2021). Attention is not all you need: Pure attention loses rank doubly exponentially with depth.
>
> [2] Ye, T. et al. (2024). Differential transformer.
>
> [3] Wortsman, M. et al. (2023). Replacing softmax with relu in vision transformers.

---

### Author Response · Authors · 2024-11-18
**General Response**

We thank the reviewers for their taking the time to review the manuscript. A few general themes emerged which we think beneficial to discuss before a point-by-point response.

- The clarity of the theoretical contributions and the simplicity of the model are listed as strengths and weaknesses of the manuscript, respectively. We have chosen our model to maximise the clarity of the theoretical analysis and its main takeaway: the softmax attention mechanism has a spectral gap, which is a key contributor to rank collapse — a topic under discussion in the machine learning community. We develop a rigorous theory and show that it is consistent with the key-query model in the first attention layer. Subsequent layers exhibit greater complexity as the details of their associated Markov matrices are unknown. Achieving analogous precise results for these layers would require developing some new random matrix theory, which lies beyond the scope of this initial investigation. Experimental results, now detailed in the updated Section 3, demonstrate that more complex architectures — including famous transformer models such as BERT — also suffer from exacerbated rank collapse due to the use of softmax attention. Let us emphasize that employing simplified models is a common approach in papers investigating the theory of deep learning, as it facilitates a rigorous analysis of the issue under study; see, for instance, [1], [2], and others. Our contribution is, in fact, at the cutting edge of the analysis of transformers at initialisation. We ask the reviewers to reconsider their scores by acknowledging the complexity and elegance of the significant theoretical contributions reflected in the appendices.

- The lack of extensive empirical validation is highlighted as a weakness by reviewers. While we agree that such an addition would give the manuscript an extra dimension, we believe that it should be addressed separately. Notably, three recent manuscripts with a more experimental focus — [3], [4], and [5] — propose solutions similar to ours. For instance, [3] empirically explores "centred attention'', which corresponds to the removal of the spectral gap, formalised in our work; see their Section 4 for more extensive experiments that validate our proposal. We have included citations to these manuscripts to further substantiate the relevance of our work and to guide readers interested in the practical implications of our theory. While we agree that our work opens exciting avenues for future empirical studies, our findings are robust, stand-alone and empirically supported within our theoretical framework. ICLR lists "theoretical issues in deep learning" as a key topic, and our work clearly aligns with this, meeting ICLR’s standards without necessarily requiring comprehensive validation across real-world tasks. We respectfully ask the reviewers to evaluate our work solely based on its scope and content, irrespective of potential personal preferences.

- Reviewers identified as a weakness that the manuscript does not consider training dynamics. While we agree that this would be a key addition to the field, it is beyond the scope of the current work. That said, we believe the theoretical contributions and the novelty of our results will appeal to a significant number of attendees at ICLR, making the manuscript deserving of acceptance. This work addresses signal propagation in transformers at initialisation, which is the subject of a sizeable and growing body of literature; see [6], [7], [1], and [2]. Please also note that our paper introduces a set of tools — random matrix theory and free probability — that have not previously been applied to the analysis of transformers. We believe these tools have significant potential to inspire and drive future theoretical developments, as they have demonstrated in the present work.


Having clarified the above common points raised by reviewers, we now discuss individual comments accordingly and invite the reviewers to reconsider their evaluation of our work considering this note.

----

[1] He, B. et al. (2023). Deep transformers without shortcuts: Modifying self-attention for faithful signal propagation.

[2] Noci, L. et al. (2022). Signal propagation in transformers: Theoretical perspectives and the role of rank collapse.

[3] Ali, A. et al. (2023). Centered self-attention layers.

[4] Noci, L. et al. (2024). The shaped transformer: Attention models in the infinite depth-and-width limit.

[5] Ye, T. et al. (2024). Differential transformer.

[6] Hron, J. et al. (2020). Infinite attention: NNGP and NTK for deep attention networks.

[7] Dong, Y. et al. (2021). Attention is not all you need: Pure attention loses rank doubly exponentially with depth.

---

### Author Response · Authors · 2024-11-24
**Willing to discuss**

We thank the reviewers for their feedback and for taking the time to evaluate our paper. We have carefully addressed all the concerns and questions raised in the reviews and provided detailed responses. We hope that our clarifications have resolved the mentioned issues.

If there are any additional questions or suggestions, we would be happy to address them promptly. We would also greatly appreciate it if the reviewers could revisit our responses and consider updating their evaluations if the clarifications have adequately addressed their concerns.

---

### Author Response · Authors · 2024-11-25
**Gentle Reminder**

We kindly remind the reviewers to please submit their feedback and update their scores by 26 November to help us refine and improve our work to the fullest extent. Thank you!

---

### Author Response · Authors · 2024-11-29
**Rebuttal Summary**

With no further responses received despite the extended discussion period, we will summarise our rebuttal once again. Our work presents a cutting-edge analysis of transformers' initialisation, utilising advanced results from random matrix theory and opening up numerous exciting avenues for future research. In most cases, the reviewers have mistaken these promising directions—such as the extension of our results to deeper layers, training dynamics, and extensive empirical validation—for weaknesses and grounds for rejecting the paper, despite not having identified any specific flaws in our presented analysis.

We have made every effort to explain why these emerging open questions not only highlight the significance of our work but also emphasise its potential for large-scale impact. They can be viewed as additional reasons to accept the paper, rather than excuses for rejection. **Will our contribution benefit the field by inspiring further advancements if published? We firmly answer in the affirmative, and we trust the Area Chairs will recognise this and make their final decision accordingly.**

---

### Meta-Review · Area_Chair_845P · 2024-12-18

**Metareview:**

a) The paper considers the issue of rank collapse in softmax attention, which can cause training instabilities. The authors identify a new variant of rank collapse: in width, which is the focus of this paper. By modeling softmax-based attention at initialization using Random Markov matrices, the paper found the root cause of the problem as the spectral gap between the two largest singular values of the attention matrices. By eliminating this spectral gap, the issue of rank collapse is resolved.

b) The paper is well organized, with a clear theoretical narrative and easy to read. The proposed analysis is simple, the found issue is interesting and the proposed solution is also simple and easy to implement.

c) The main weakness of the paper is the lack of empirical validation. Author provided a comparison of training loss, however, it is important to validate the theoretical finding on actual transformer models in either language or vision tasks and measure what is the impact of the issue and the found solution on the final tasks. In addition the proposed analysis is simplified, based on an attention-only single-head transformer. What would happen on more realistic configurations is not analyzed, nor theoretically (which can be hard) neither empirically.

d) The main reason for rejection are the weaknesses exposed above. The paper is well written and easy to follow, however, without an empirical validation of the proposed solution to contrast rank collapse in width, it is difficult to estimate the real impact that this contribution can have on the research community. Therefore, the theoretical contribution, without empirical evidence is not enough for acceptance especially considering that the proposed analysis has many simplifications which make it far from real setting.

**Additional Comments On Reviewer Discussion:**

Rev. ALfq provided a review in which among other points he raised the issue of lack of experimental evaluation. Unfortunately rev. did not engage in a discussion with the authors, so it is not clear if the provided answers satisfied them or not. The final score is 3. Although to me it is too low for the raised critical points, I agree about some of the issues to which authors did not provide a satisfactory answer, specifically in the empirical evaluation.

Rev. Ua2p consider the work of highly theoretical significance and assigned a score of 6, while still pointing out the lack of empirical validation to assess when and how the proposed solution is useful in real problems.

Rev. Htgk assigned a score of 5 to the paper, appreciating the theoretical contribution but again pointing out the simplified assumptions among other weaknesses. They did not acknowledge the authors answer.

Rev. smfh also assigned a score of 5, pointing out the limitations of the proposed theoretical analysis, such as the limited validity to the first layer and the unclear understanding of the proposed solution with more than two layers. Authors provided a rebuttal, with some interesting directions to improve the validity of their results, but for the rev. that was not enough to change their opinion.

Overall, all reviewers had a similar understanding of the paper: nice theoretical contribution but not clear applicability in practice due to oversimplified assumptions and lack of theoretical validation. While theoretical contributions are well appreciated, as those theories are applied to the neural network training, their significance is relevant only when it is matched with a clear applicability. Here, this is not the case due to over simplifying assumptions in the analysis and lack of empirical validation. Instead of rejecting the suggestion of an empirical evaluation, authors should have tried to provide some evidence to compensate the limitations of the analysis.

---

### Decision · Program_Chairs · 2025-01-22

Reject